# *mmsig*: a fitting approach to accurately identify somatic mutational signatures in hematological malignancies

Even H. Rustad[1,2], Ferran Nadeu [3,4], Nicos Angelopoulos[5], Bachisio Ziccheddu[6,7], Niccolò Bolli[8,9],
Xose S. Puente [10], Elias Campo [3,4,11], Ola Landgren [1,7✉] & Francesco Maura [1,7✉]

Mutational signatures have emerged as powerful biomarkers in cancer patients, with prognostic and therapeutic implications. Wider clinical utility requires access to reproducible algorithms, which allow characterization of mutational signatures in a given tumor sample. Here, we show how mutational signature fitting can be applied to hematological cancer genomes to identify biologically and clinically important mutational processes, highlighting the importance of careful interpretation in light of biological knowledge. Our newly released R package *mmsig* comes with a dynamic error-suppression procedure that improves specificity in important clinical and biological settings. In particular, *mmsig* allows accurate detection of mutational signatures with low abundance, such as those introduced by APOBEC cytidine deaminases. This is particularly important in the most recent mutational signature reference (COSMIC v3.1) where each signature is more clearly defined. Our mutational signature fitting algorithm *mmsig* is a robust tool that can be implemented immediately in the clinic.

[1] Myeloma Service, Department of Medicine, Memorial Sloan Kettering Cancer Center, New York, NY, USA. [2] Institute for Cancer Research, Oslo University Hospital Radiumhospitalet, Oslo, Norway. [3] Patologia Molecular de Neoplàsies Limfoides, Institut d'Investigacions Biomèdiques August Pi i Sunyer (IDIBAPS), Barcelona, Spain. [4] Centro de Investigación Biomédica en Red de Cáncer (CIBERONC), Madrid, Spain. [5] School of Medicine, Systems Immunity Research Institute, Cardiff University, Cardiff, UK. [6] Department of Molecular Biotechnologies and Health Sciences, University of Turin, Turin, Italy. [7] Myeloma Program, Sylvester Comprehensive Cancer Center, University of Miami, Miami, FL, USA. [8] Department of Oncology and Hemato-Oncology, University of Milan, Milan, Italy. [9] Unità Operativa Complessa di Ematologia, Fondazione IRCCS Ca' Granda Ospedale Maggiore Policlinico, Milan, Italy. [10] Departamento de Bioquimica y Biologia Molecular, Instituto Universitario de Oncología (IUOPA), Universidad de Oviedo, Oviedo, Spain. [11] Unitat Hematopatologia, Hospital Clínic of Barcelona, Universitat de Barcelona, Barcelona, Spain. ✉email: col15@miami.edu; fxm557@med.miami.edu

The mutational profile of a tumor represents an archive of all mutational processes that have been active throughout tumor phylogeny, dating back to the fertilized egg[1,2]. Analyzing the mutational signatures associated with distinct mutagenic processes has revealed key insights into cancer pathogenesis, evolution, and potential therapeutic strategies[1,3–6]. To make up a mutational profile, single nucleotide variants (SNVs) are commonly divided in 96 classes based on their trinucleotide context, defined by the mutated base and the most proximal bases in the 5′ and 3′ direction[1,3]. Recent whole-genome sequencing (WGS) studies have revealed more than 40 signatures of single base substitutions (SBS), representing processes active in all tissues (e.g., ageing related clock-like processes), as well as cell-type-specific intrinsic processes (e.g., APOBEC-family cytidine deaminases, such as activation-induced cytidine deaminase, AID), processes related to exogenous agents (e.g., smoking and chemotherapeutic agents), and deficiencies of specific DNA repair mechanisms (e.g., homologous repair deficiency, HRD)[1–3,7–12]. Several tools for mutational signature analysis have been proposed, without a single accepted standard having emerged. Resulting from the lack of methodological consensus, different studies have often reported conflicting and biologically implausible results[10,11,13,14].

As a first step toward standardization of mutational signature analysis in hematological malignancies, we recently proposed a three-step workflow of (1) de novo signature extraction (e.g., by non-negative matrix factorization, NMF); (2) assignment to a set of reference signatures (e.g., COSMIC v3.1; https://cancer.sanger.ac.uk/cosmic/signatures/); and (3) applying a fitting algorithm to determine which of the signatures defined in steps 1–2 are present in each tumor[10]. We previously focused on the challenges and pitfalls of the first two steps, which may result in falsely identifying signatures that are not active in a given disease, such as HRD-SBS3 in multiple myeloma (MM)[10,14]. Successfully applying the most recent de novo extraction tools and assignment of mutational signatures to a large set of samples with a specific cancer will reveal the main mutational processes active in that disease[2]. However, as previously reported, this approach may be affected by several issues leading to incorrect quantification of signature contribution[10]. One such problem is inter-bleeding of signatures, where a mutational signature present in only a subset of cases are incorrectly called in the others as well. Furthermore, de novo extraction requires large cohorts of patients, limiting their applicability in the clinic. In cancers with well-characterized mutational signature landscapes, fitting algorithms are highly suited for clinical application, as they can be applied to individual samples with short run-times[5,12,15,16]. Although technically straightforward to perform, mutational signature fitting requires rigorous interpretation to ensure the accuracy and validity of results.

Here, we show how mutational signature fitting can be used to identify clinically relevant mutational processes in hematological malignancies. We highlight the differences between commonly used algorithms and introduce the novel algorithm *mmsig*, developed specifically to solve difficult cases where it is unclear whether a given signature is present or absent.

## Results

**Mutational signature fitting and *mmsig*.** In principle, mutational signature fitting is a mathematical procedure aimed to determine the combination of known signatures that best explains the observed mutational profile (Fig. 1A, B)[3]. This is often measured by the cosine similarity of the original mutational profile with the profile that is reconstructed based on the fitted signatures (reconstruction accuracy; Fig. 1C, D)[3,16]. However,

from a practical perspective, it is most important to identify those signatures that have a clear biological correlate and may serve as clinical biomarkers. When algorithms are developed with optimal reconstruction accuracy as the principal goal, this often comes at the cost of overfitting, resulting in false positive signature calls[10]. We developed the novel algorithm *mmsig* with this in mind to optimize specificity at a minimal loss of reconstruction accuracy ("Methods").

To illustrate the challenges of mutational signature fitting and the unique features of *mmsig*, we analyzed a thoroughly characterized WGS dataset with 82 MM samples[11,17–19]. We included the 8 mutational signatures previously identified in MM: SBS1 and SBS5, mutations related to the cell aging (i.e., clock-like); SBS2 and SBS13, resulting from APOBEC cytidine deaminase activity; SBS9, attributed to the non-canonical genome-wide action of AID (nc-AID); SBS8 of unknown etiology; SBS18, which may be related to DNA damage from reactive oxygen species; and SBS-MM1, the mutational footprint of melphalan therapy (Fig. 1B)[11]. A subset of patients with MM show mutational profiles dominated by APOBEC mutagenesis, with high proportion of mutations in TCT and TCA context that are well-explained by SBS2 (C>T) and SBS13 (C>G) (e.g., patient PD26419a in Fig. 1A). Patients with lower APOBEC were dominated by SBS5, with distinctive peaks in T>G attributable to SBS9 (e.g., patient V0D58T in Fig. 1A).

Using the latest COSMIC reference (i.e., COSMIC v3.1; https://cancer.sanger.ac.uk/cosmic/signatures/SBS/index.tt) and applying three different fitting algorithms (*deconstructSigs*, *mutationalPatterns*, and *mmsig*) to each of the 82 MM samples without the use of error-correction showed similar results (Fig. 2A). Accordingly, pairwise comparisons of the reconstructed mutational profile for each sample by the different algorithms showed median cosine similarity of >0.998. Moreover, the median cosine similarity between the reconstructed and original mutational profiles was 0.97 for all three algorithms. However, when the fitted mutational signatures were compared with actual exposure of patients to melphalan, it became clear that SBS-MM1 was falsely identified in >90% of samples from untreated patients. This problem of overfitting is well known, and a common approach to avoid false positives is to remove all signatures with estimated contribution below a given percentage[15,16]. A threshold of 6% is the standard setting for *deconstructSigs*, and we applied the same threshold to *mutationalPatterns* as well. Although removing false positive SBS-MM1 in most cases, this approach introduced problems of its own, because the COSMIC v3.1 versions of APOBEC and SBS1 often fell below the 6% threshold despite being clearly present (Fig. 2B). SBS1 has been identified in all cancers and normal individuals, and APOBEC is known to be active in the vast majority of patients with MM[7,11,20–23]. Moreover, the APOBEC signatures have highly distinct profiles, making it possible to identify them at low contributions even by visual inspection of the mutational profile (see Fig. S1). The error-correction approach built into *mmsig* takes advantage of the distinctiveness of each mutational signature, by keeping in the final output only those signatures that lead to considerably lower reconstruction accuracy when removed (>1% reduction in cosine similarity is the standard setting). Because SBS1 and SBS5 are known to always be present, we kept them in the final profile no matter what their contribution was to the overall reconstruction accuracy. The results of all three algorithms after error-correction can be seen in Fig. 2B, along with their reconstruction accuracy before and after error-correction (Fig. 2C, D). In contrast to the other algorithms, *mmsig* with error-correction was able to correctly identify APOBEC in the majority of samples while avoiding false positive SBS-MM1, at the cost of a 0.005 reduction in median reconstruction accuracy.

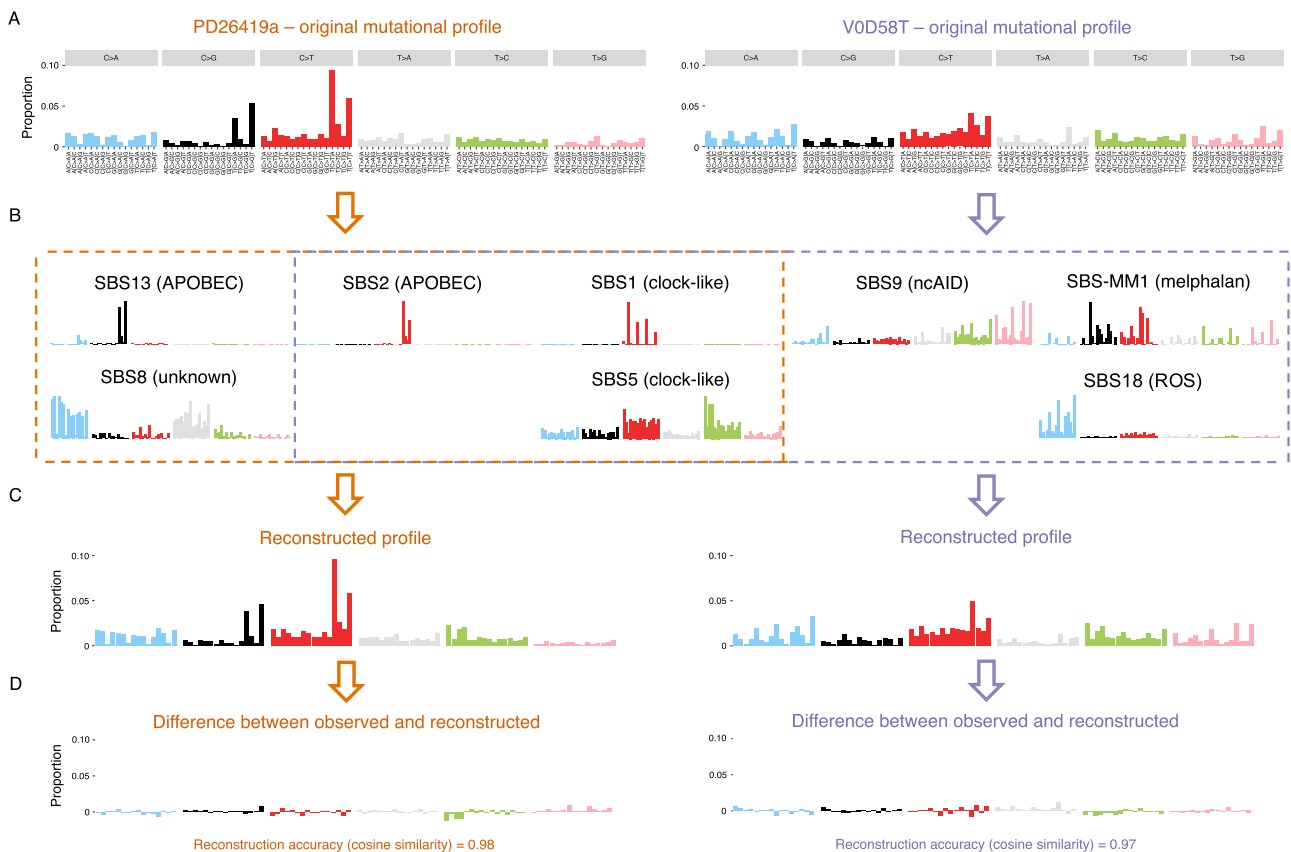

**Fig. 1 Mutational fitting in multiple myeloma. A** Mutational profiles from two example patients with MM. **B** Mutational signature fitting reveals the main APOBEC signature (SBS2) along with the clock-like signatures SBS1 and SBS5 in both samples. Sample PD26419a (left) also had evidence of SBS13, which is most often detected in patients with high APOBEC mutational burden, as well as SBS8, a common mutational signature in MM that is of unknown etiology. Sample V0D58T (right) showed contributions from non-canonical activation-induced cytidine deaminase (nc-AID; SBS9), melphalan-induced mutagenesis (SBS-MM1), and damage by reactive oxygen species (ROS; SBS18). **C** To measure how well the fitted signatures account for the actual mutational profile in each sample, the first step is to reconstruct a mutational profile by multiplying the weight assigned to each signature with the reference profile of that signature. **D** Subtracting the reconstructed profile from the original profile illustrates which parts of the mutational spectrum are well-explained and may point to the presence of additional signatures that were not included in the analysis. In these cases, the mutational profiles were well-explained by the fitted signatures (high reconstruction accuracy), indicated by cosine similarity of >0.97 between observed and reconstructed profiles.

**Importance of the reference signatures**. Since the concept of mutational signatures was first introduced in 2012, the most commonly used signature reference has been updated from the initial 30 COSMIC signatures (https://cancer.sanger.ac.uk/cosmic/signatures_v2.tt) (i.e., COSMIC v2)[1,4] to a new catalog of 49 reference signatures (i.e., COSMIC v3.1)[2], with additional signatures having been reported as the consequence of exogenous agents[9,11,24]. Importantly, the updated COSMIC reference led to sharper signature definitions, removing background noise and contamination from other signatures. For example, the mutational signature first reported as SBS1 included a flat background similar to SBS5 that has been removed in the new version, leaving only the characteristic and biologically accurate C>T in CpG mutations (see Fig. S2A)[2,25]. Moreover, the APOBEC mutational signature (i.e., SBS2 and SBS13) profiles showed a degree of overlap in the previous reference and have since been clearly separated into one C>T signature (SBS2) and one C>G signature (SBS13) (see Fig. S2B)[2,26]. To illustrate the effects of these changes on mutational signature fitting, we repeated the above analysis of 82 MM samples using the original COSMIC v2 reference with the addition of SBS-MM1. The estimated contribution of each signature was significantly different depending on which reference was used (Fig. 3A). As expected, the estimated SBS1 contribution was higher using the old reference (Fig. 3B), as

were the estimates for SBS2, SBS8, and SBS9. A decrease in SBS5 was seen with the old reference, most likely reciprocal to the presence of a flat "background" in several other mutational signatures (Fig. 3C). Conversely, SBS18 was considerably higher with the new reference, as was SBS-MM1. This latter point is important because the SBS-MM1 reference was the same in both analyses, but the assignment of mutations to SBS-MM1 was affected by the changes to other signature references. As expected, updating to a reference set where each mutational signature is more clearly defined resulted in a clear overall increase in reconstruction accuracy for all three fitting algorithms (Fig. 3D). Applying the same filters as described above resulted in lower reconstruction accuracy, with similar performance for all three algorithms (Fig. 3E). This is in contrast to results from the new reference, where *mmsig* performed considerably better (Fig. 2D). We note that *deconstructSigs* and *mutationalPatterns* were first released before the latest mutational signature reference was released and thus optimized for the less specific COSMIC v2 reference. The iterative filtering approach of *mmsig*, based on cosine similarity rather than a hard cut-off, may be particularly advantageous in identifying mutational signatures that are highly distinctive but account for a low proportion of the mutational profile (e.g. APOBEC and SBS1). Moreover, irrespective of which mutational signature reference was used, only *mmsig* could avoid

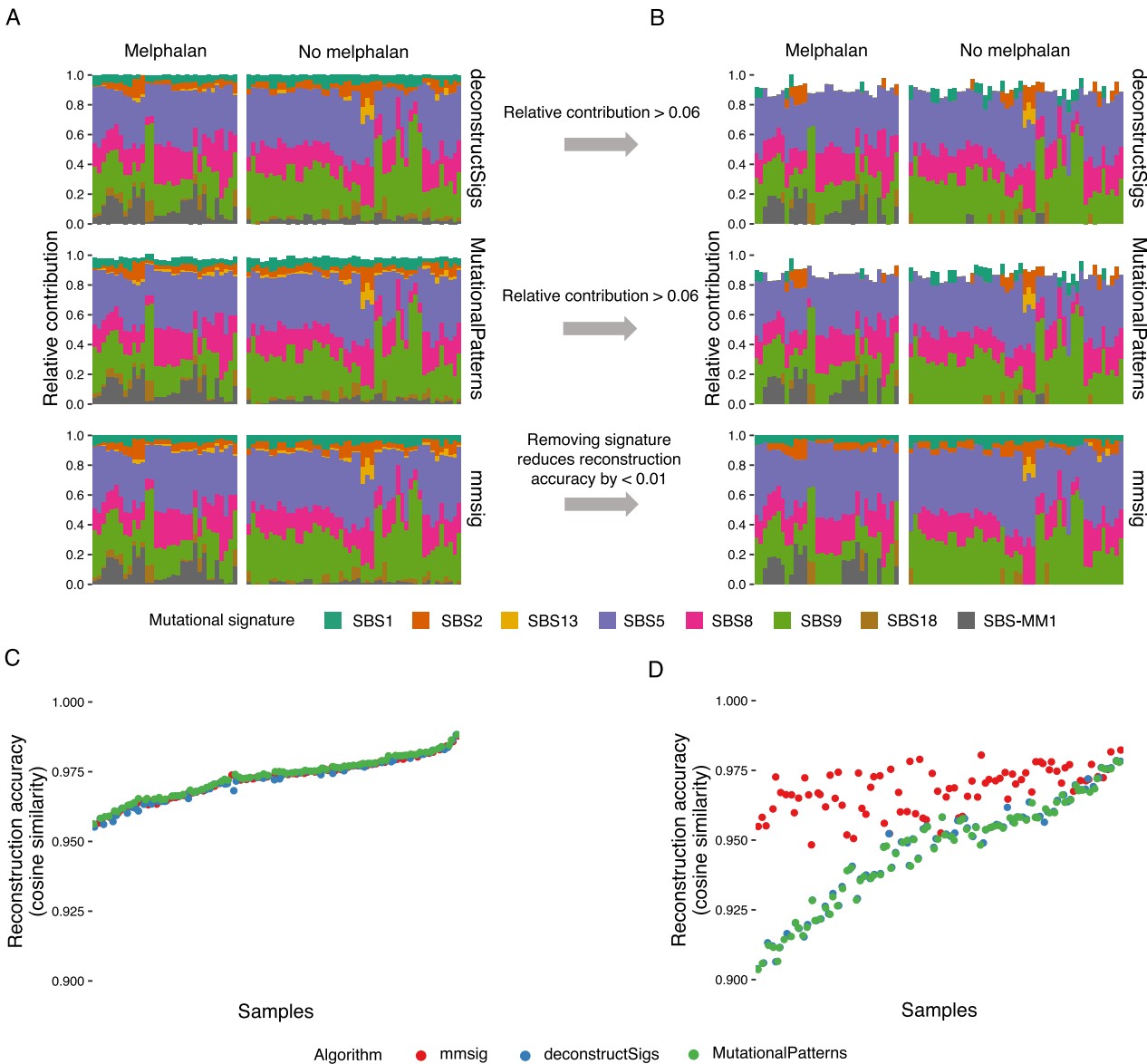

**Fig. 2 Mutational signature fitting with *mmsig* shows high sensitivity, specificity, and reconstruction accuracy. A** Without any filtering procedures, the results from *deconstructSigs*, *mutationalPatterns*, and our novel algorithm *mmsig* are highly similar. Of note, all of the algorithms falsely identify the mutational signature of melphalan-induced mutagenesis (i.e., SBS-MM1) in >90% of patients who were not exposed to melphalan, representing false positives. **B** After applying the recommended filtering procedures, *mmsig* shows 100% specificity for SBS-MM1 while retaining the contributions from SBS1 and SBS2. Filtering of output from *deconstructSigs* and *mutationalPattern* is less flexible, relying on a relative contribution threshold, which results in false negative SBS1 and SBS2 while retaining some cases of false positive SBS-MM1. **C** The reconstruction accuracy of *deconstructSigs*, *mutationalPatterns*, and *mmsig* are virtually identical when no filtering is applied. **D** When filtering procedures are applied, *mmsig* shows superior reconstruction accuracy >0.95 for all samples. Filtered output from *deconstructSigs* and *mutationalPatterns* was scaled for the sum of mutational processes to equal 1.

false positive SBS-MM1 in patients without prior melphalan exposure (see Fig. S3).

**Resolving uncertainty in mutational signature fitting.** When it is particularly important to determine if a mutational process is active, two additional steps can be taken to increase specificity ("Methods"). First, 95% confidence intervals (CI) for each mutational signature estimate can be constructed by resampling with replacement from the original mutational profile of a sample. Second, transcriptional strand bias can be considered for signatures associated with transcription-coupled repair[2,24]. If the lower bound of the 95% CI for a mutational signature is above zero, and the typical transcriptional strand bias is present, it is

highly likely that the signature is actually present. Below, we illustrate this approach in settings where we have prior knowledge of which signatures are present in each sample: chemotherapy-related signatures in MM and acute myeloid leukemia (AML), and nc-AID activity in chronic lymphocytic leukemia (CLL). Having established above that *mutationalPatterns* and *deconstructSigs* provide near-identical results, we moved forward comparing *mmsig* with *deconstructSigs* alone.

First, we confirmed the presence of SBS-MM1 in melphalan-exposed patients with MM (Fig. 4A and Fig. S4). In all cases with >10% estimated signature contribution, the presence of SBS-MM1 was stable in the face of random resampling of mutations (i.e., the CI clearly did not include zero). In all but one of these samples, there was also evidence of transcriptional strand bias in C>T

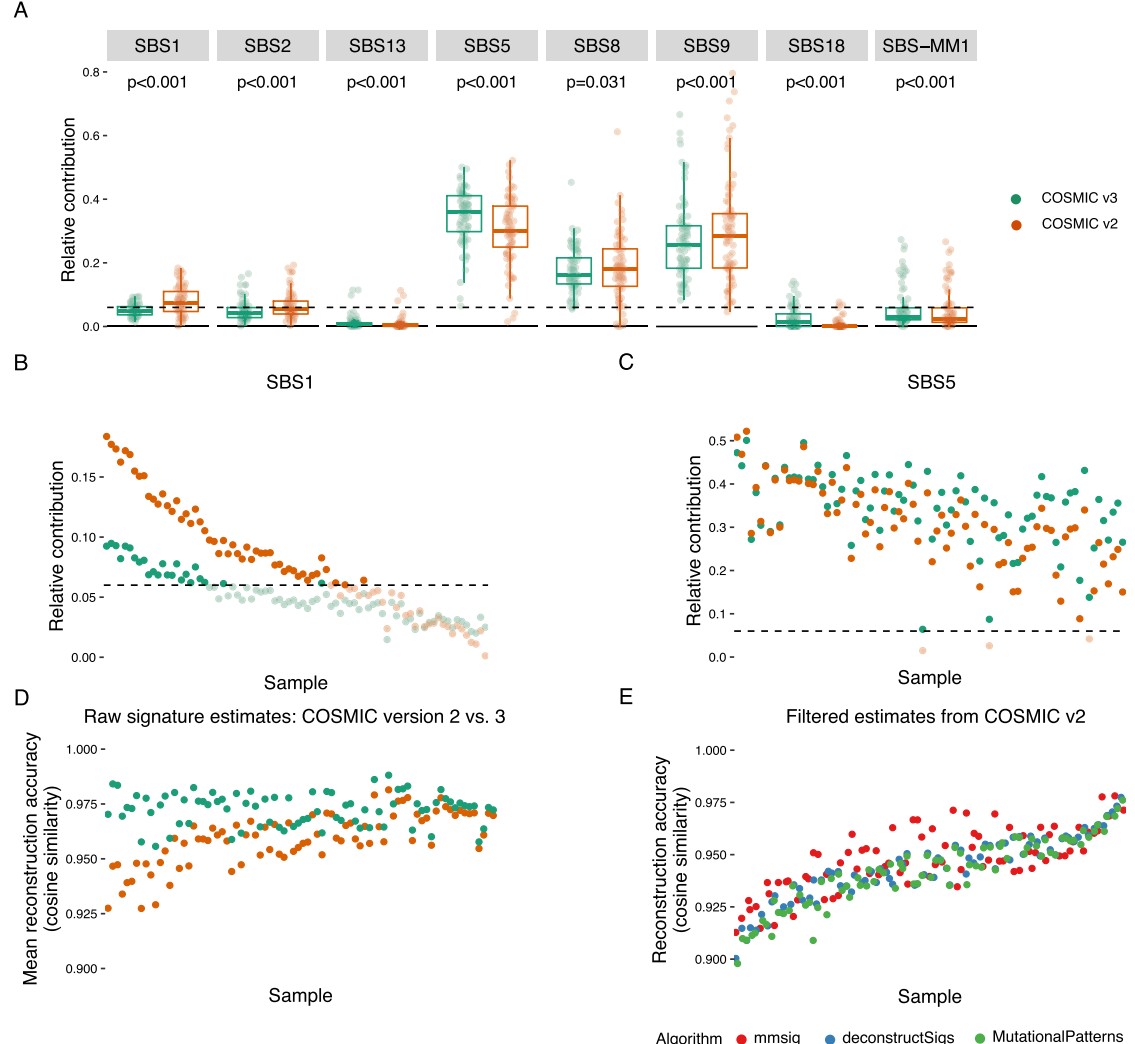

**Fig. 3 The choice of mutational signature reference version significantly impacts the performance of fitting algorithms.** Unfiltered mutational signature contributions for each sample were determined as the mean of estimates from *deconstructSigs*, *mutationalPatterns*, and *mmsig* (the three algorithms produced almost identical results). **A** The mutational signature contributions were significantly different depending on whether the original COSMIC v2 reference or the latest updated COSMIC v3.1 from Alexandrov et al. was used for fitting[1,2]. Boxplots show the relative contribution of each signature in each sample, displaying the median and interquartile range with outlier samples drawn as dots. *P*-values were estimated by paired Wilcoxon tests. Dashed black lines mark the 6% relative contribution threshold used for filtering of results from *deconstructSigs* and *mutationalPatterns*. **B**, **C** Showing individual sample data from **A** for SBS1 and SBS5. **B** Estimates for SBS1 were generally higher using the old reference, where SBS1 contains "contamination" of a flat background that may be better explained by SBS5. Notably, this resulted in SBS1 ending up above the 6% filtering threshold (dashed black line) in a higher fraction of samples using the old reference. **C** Estimates of SBS5 were generally higher using the new reference. **D** The overall cleaner mutational signatures in the new reference resulted in higher reconstruction accuracy compared with when the old reference was used. **E** When using the COSMIC v.2 reference, the reconstruction accuracy was similar irrespective of which fitting algorithm was used: *deconstructSigs*, *mutationalPatterns* or *mmsig*. This is in contrast to the new reference, where *mmsig* was superior.

mutations across the trinucleotide contexts typically associated with SBS-MM1: CCA, GCA, GCC, GCG, and GCT. Not all patients previously exposed to melphalan had detectable SBS-MM1, because this requires a single melphalan-exposed cell to be positively selected and expand beyond the limit of detection of WGS, which does not always occur[11,24,27]. An alternative explanation is the engraftment model, where myeloma cells are infused during autologous stem cell transplantation, thus avoiding melphalan exposure[27]. There will also be a range of exposures (e.g., <10%) where there is insufficient evidence to ascertain whether SBS-MM1 is really present (i.e., transcriptional strand bias and CIs). In such ambiguous cases, SBS-MM1 may not be essential to explain the overall mutational profile, leading to SBS-MM1 being removed from the final profile by the error-suppression

procedure of *mmsig*. We prefer to be conservative in these cases and tolerate false negatives to a geater degree than false positives. In patients without prior melphalan exposure, SBS-MM1 with its characteristic transcriptional strand bias was consistently absent (Fig. 4A).

Platinum-based chemotherapy is the underlying cause of mutational signature SBS35[2,24]. Here, we applied *deconstructSigs* and *mmsig* with standard cutoffs to two patients with therapy-related AML previously exposed to platinum-based chemotherapy as well as 47 patients with de novo AML (platinum naive) (Fig. 4B). Point estimates of the SBS35 contribution (i.e., standard mutational signature fitting) identified SBS35 in both patients with prior platinum exposure. Strikingly, the false positive rate for SBS35 among de novo AML cases was 21% for *mmsig* and 70%

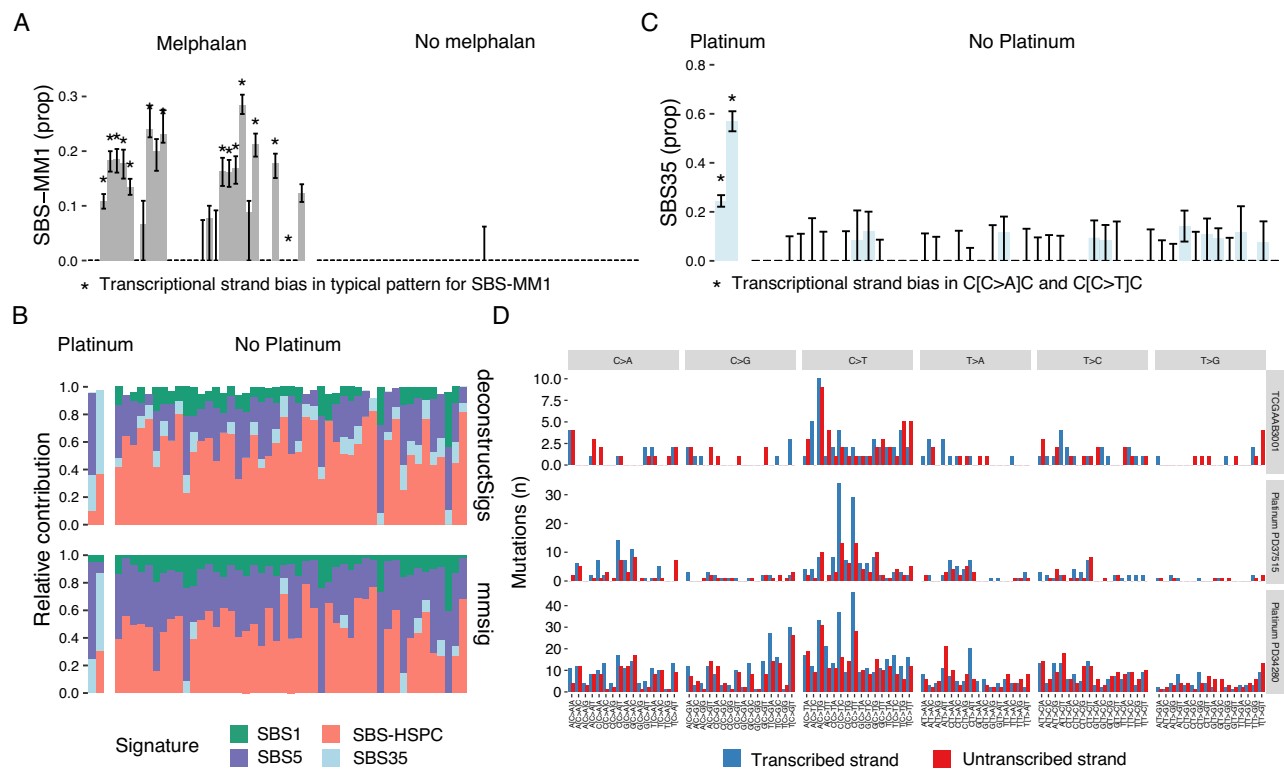

**Fig. 4 Resolving uncertainty in mutational signature fitting. A** Bar chart showing the estimated SBS-MM1 contribution in 82 MM samples with and without prior melphalan exposure by *mmsig* (similar to Fig. 2B), with 95% CI estimated from 1000 bootstrapping iterations. Asterisks indicate statistically significant transcriptional strand bias in characteristic trinucleotide contexts (Poisson test $p < 0.05$). **B** Stached bar chart showing mutational signature profiles of 49 patients with acute myeloid leukemia (AML) with ($n = 2$, left) and without ($n = 47$, right) prior platinum exposure, estimated using *deconstructSigs* (top) and *mmsig* (bottom). SBS-HSPC is a mutational signature characteristic of hematopoietic stem cells. **C** Similar to A for 49 AML patients with and without prior platinum exposure. **D** Bar chart showing the number of mutations on the transcribed (blue) and untranscribed (red) strand in each trinucleotide context for three patients. On the top we show one patient who did not have prior history of platinum exposure, but where SBS35 was identified with the whole 95% CI above zero. The mutational signature profile does not fit the classic pattern of SBS35, however, and there is no sign of the characteristic transcriptional strand bias, suggesting a false positive call. The two lower panels show the patients with therapy-related AML who were exposed to platinum. Their mutational profiles are highly dominated by SBS35 with strong transcriptional strand bias.

for *deconstructSigs*. This highlights the importance of additional measures to improve specificity. Presence of SBS35 was confirmed with high confidence in both of the platinum-exposed patients (i.e., non-zero 95% CI and strong transcriptional strand bias in C[C>A]C and C[C>T]C); but in none of the de novo AML cases (Fig. 4C). In one de novo AML sample (TCGAAB3001) the 95% CI was above zero (estimated SBS35 contribution 14.1%; 95% CI 7.8–20.5%), but the characteristic transcriptional strand bias was always absent. In ambiguous cases like this, it can be helpful to visually review the 96-class mutational profile for presence of the key features of the mutational signature in question. As illustrated in Fig. 4D, sample TCGAAB3001 lacked the characteristic peaks of SBS35 in both C>A and C>T (the tall peak in A[C>T]G is most likely attributable to the age-related deamination of 5-methylcytosine to thymine, i.e., SBS1). This is in contrast to the two samples with known platinum exposure, showing the classical profile with dominant peaks in C>T with mutations strongly favoring the transcribed strand.

CLL can be classified based on the presence or absence of somatic hypermutation of the immunoglobulin heavy chain variable region (*IGHV*) (threshold at >2% mutated bases), where unmutated *IGHV* is strongly associated with poor outcomes[2,10,28–30]. We have previously shown, using mutational signature extraction algorithms, that mutated *IGHV* status is associated with the genome-wide footprint of nc-AID activity (i.e., SBS9), whereas the unmutated subgroup lacks this mutational process[10,31]. Here, we analyzed

whole-genome sequencing data from 142 patients with CLL with mutated ($n = 68$) or unmutated ($n = 74$) *IGHV* gene. Using *mmsig* with standard filters, three patients with unmutated CLL had evidence of SBS9 (Fig. 5A); one of which also had a non-zero CI (CLL1078; Fig. 5B). This sample had an estimated SBS9 contribution of 30% (95% CI 27–34%) and 1.7% of the bases in *IGHV* were mutated by Sanger sequencing. Unmutated IGHV status was confirmed by the novel algorithm *IgCaller* applied to WGS data[32]. This patient had been classified as memory-CLL based on the epigenetic profile, consistent with having passed through the germinal center[30]. Visual inspection of the mutational signature profile did indeed show the characteristic profile of SBS9, indicating that considerable genome-wide nc-AID activity can be observed despite relatively few coding mutations in the *IGHV* region (Fig. 5C and Fig. S5)[10]. It remains to be seen whether the degree of genome-wide nc-AID activity may further refine the prognostic classification of CLL.

Our observations thus far have suggested that *mmsig* is able to confidently identify highly characteristic mutational signatures such as SBS2 at low abundance, whereas the sensitivity is slightly lower for less distinctive signatures. To quantify this effect, we simulated a series of MM genomes based on the median contribution of each mutational signature observed in WGS data. Starting with such an average genome as background, we estimated the sensitivity of each algorithm to detect signatures at a given admixture (Fig. 6A, B). In concordance with our

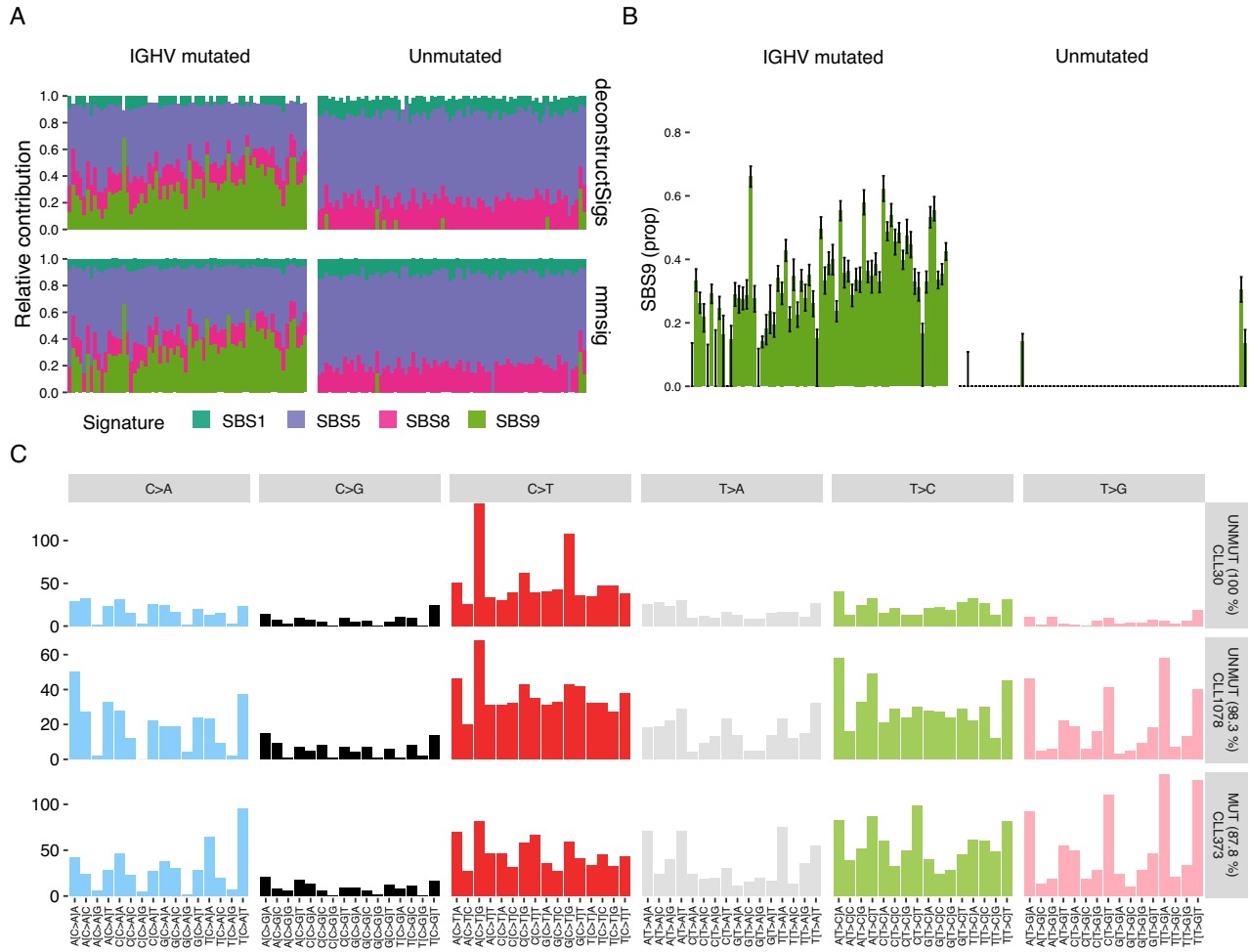

**Fig. 5 Identifying non-canonical AID-induced mutagenesis in chronic lymphocytic leukemia. A** Mutational signature profiles of 142 patients with chronic lymphocytic leukemia (CLL) with mutated ($n = 68$, left) or unmutated ($n = 74$, right) *IGHV* gene, estimated using *deconstructSigs* (top) and *mmsig* (bottom). **B** Bar chart showing the estimated proportion of mutations caused by non-canonical AID (nc-AID; SBS9) with 95% CI generated by 1000 bootstrapping iterations. **C** 96-class mutational profiles of three patients from top to bottom: unmutated CLL without evidence of nc-AID activity; unmutated CLL based on *IGHV* identity (<2% mutations), but evidence of genome-wide nc-AID activity; CLL with highly mutated *IGHV* as well as strong nc-AID signature genome-wide.

observations from real data, all three algorithms showed virtually identical results when no filters were applied. After applying standard filtering, the sensitivity of *mutationalPatterns* and *deconstructSigs* centered around 6% contribution, with >95% sensitivity achieved at 7–8% contribution. For *mmsig*, given the unique features of its error-correction approach, the sensitivity varied between mutational signatures. The sensitivity was higher for distinctive mutational signatures such as SBS2, with 100% sensitivity achieved at 4% contribution (Fig. 6A), and lower for less well-defined signatures such as SBS-MM1, with >95% sensitivity achieved at 12% contribution (Fig. 6B). Next, we added increasing levels of noise to the simulated MM genomes and estimated the effects on mutational signature estimates (Fig. 6C). The results show that SBS-MM1, SBS5, SBS8, and SBS9 are accurately estimated by all algorithms in the absence of noise, with increasing over-estimation of particularly SBS5 by all algorithms with increasing background noise. As expected, decreasing the number of mutations also increases variability; this was also observed in real data. When the number of mutations fell below 250, there was a tendency for *mmsig* to over-estimate SBS5, most likely because this signature was always kept in the final profile irrespective of random variations in the data, which may lead to dropout of other mutational signatures. For

SBS1, given its highly distinctive profile, the estimates by *mmsig* were highly accurate irrespective of added noise and the number of mutations; this was also the case for SBS2. Finally, using simulated genomes, we confirmed that the 95% CI estimated by *mmsig* contained the pre-determined contribution of that mutational signature ~94% of the time.

**Mutational signature fitting with low mutational burden.** Until this point, we have addressed mutational signature fitting of complete mutational catalogs derived from WGS, consisting of thousands of mutations. However, there are many situations where it is desirable to perform mutational signature fitting with much smaller numbers of mutations; whether for whole-exome or targeted sequencing data, branches in a phylogenetic tree or other applications[11,23,27,33,34].

To systematically evaluate the accuracy of mutational signature fitting in real data as the number of mutations decreases, we generated sets of progressively fewer mutations by random downsampling of the WGS data presented previously and estimated signature contributions using *mmsig*. Reducing the number of mutations had the main effect of increasing variability in the mutational signature estimates, as illustrated here in two

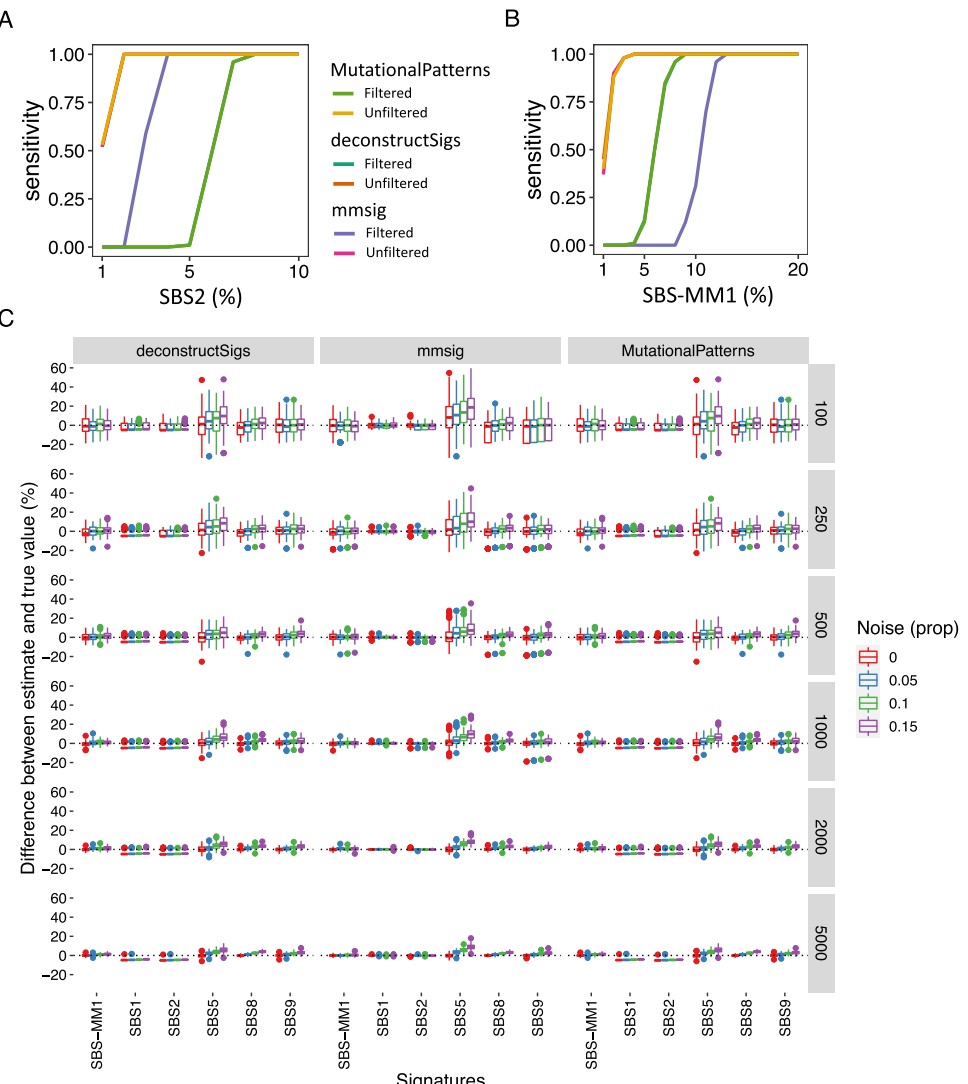

**Fig. 6 Simulated multiple myeloma genomes.** Mutational signature contributions were estimated for simulated mutational catalogs corresponding to average multiple myeloma (MM) genomes. **A**, **B** Dilution series of 1–10% SBS2 (**A**) and 1–20% SBS-MM1 (**B**) contribution in a background similar to an average MM genome. 100 simulations of 5000 mutations were performed at each dilution. Mutational signature fitting was performed by *mmsig*, *mutationalPatterns*, and *deconstructSigs* without filters and using standard filters. The lines corresponding to the sensitivity without filters are superimposed for all three algorithms consistent with virtually identical results. Regarding the results after filtering, *mutationalPatterns* and *deconstructSigs* showed virtually identical performance, leading to the sensitivity curves being superimposed, whereas *mmsig* showed either higher (**A**) or lower (**B**) sensitivity than the other algorithms. **C** 100 simulations were generated of 5000, 2000, 1000, 500, 250, and 100 mutations (rows) and mutational signature fitting performed by *mmsig, mutationalPatterns* and *deconstructSigs* with standard filters (columns). Boxplots show median and interquartile range with outliers drawn as dots. Each data point represents the estimated signature contribution in one simulated mutation catalog.

example patients (Fig. 7A). Of note, variability markedly increased below 500 mutations, disproportionately affecting flat mutational signatures such as SBS5, while the APOBEC-signatures and SBS1 were relatively spared. The mean contribution of each signature also changed slightly, which may be explained by the fact that SBS5 and SBS1 are always included in the final signature profile from *mmsig*, while other mutational signatures may disappear in some mutation sets due to random variation. Consequently, the observed means of SBS5 and SBS1 were slightly higher and the means for other signatures slightly lower than what was estimated from the full mutational catalog.

We went on to estimate how the sensitivity and specificity of the mutational signature fitting is affected by the variability of mutational profiles due to random sampling as the number of mutations decreases. Three mutational signatures were selected as

illustrative examples: SBS2 and SBS-MM1 in MM and SBS9 in CLL, each with a distinct mutational profile as well as different abundance when present (mean contribution of 6.1%, 18.7%, and 35.6%, respectively) ("Methods"). Strikingly, all of the three mutational signatures could be identified with high sensitivity and specificity all the way down to sets of 100 mutations using *mmsig* with standard settings, requiring a non-zero estimated contribution for a positive signature call (Fig. 7B). With sets of 500 or more mutations, the results were similar to those obtained from the full WGS data. Applying a more stringent criterion of non-zero 95% CI resulted in ~100% specificity across the board, but at a considerable loss of sensitivity, particularly with lower mutation counts.

Finally, we compared mutational signature fitting in the full WGS catalogs with subsets of mutations corresponding to whole-

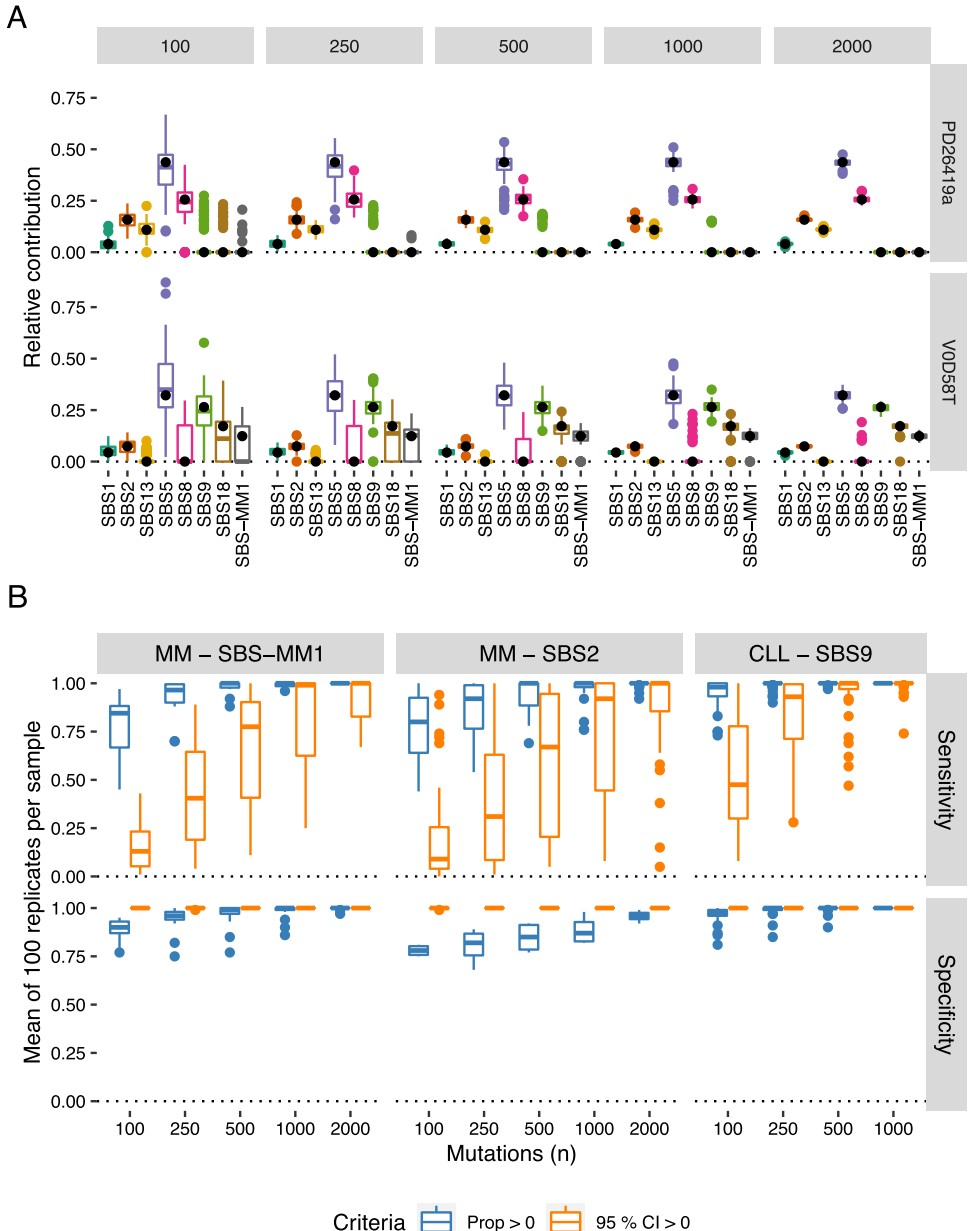

**Fig. 7 Mutational signature fitting with low numbers of mutations.** Mutational signature contributions were estimated for each patient in 100 randomly drawn catalogs of 2000, 1000, 500, 250, and 100 mutations. Boxplots show median and interquartile range with outliers drawn as dots. **A** Mutational signature profiles of two example patients (rows), showing the estimated mutational signature contributions (columns). Each data point represents the contribution of a given signature in one randomly drawn mutation set. **B** Sensitivity and specificity of *mmsig* to identify SBS-MM1 in MM (left), SBS2 in MM (middle) and SBS9 in CLL (right) when different criteria were applied (color legend). Each data point represents the mean estimated sensitivity or specificity across 100 replicates for a given patient and mutation count.

exome and targeted sequencing capture kits ("Methods"; Fig. 8). Going from WGS to WES reduced the mean number of mutations from 5437 to 245 per sample for MM and 2391 to 108 for CLL. Targeted capture panels did not yield sufficient mutation counts to proceed with mutational signature analysis. The mean numbers of mutations per sample within the capture regions were 20 for MM and 1 for CLL, with a subset of patients lacking mutations altogether (2.4% for MM and 56% for CLL). Exome-based analysis of SBS-MM1 showed excellent performance compared with WGS, consistent with results from random downsampling of WGS data (Fig. 8C). However, exome-based analysis of SBS9 in CLL showed considerably lower concordance with WGS than expected from our random

downsampling analysis. This observation can be explained by the known behavior of the SBS9 (nc-AID) mutational process, which is predominantly active in the non-coding regions of the genome[11]. Indeed, exome-based estimates in our data were consistently lower than paired WGS-based estimates (mean 12%; paired *T*-test, $p < 0.001$). Enrichment of non-synonymous mutations has been described for the APOBEC mutational signatures (i.e., SBS2 and SBS13)[11], but the overall APOBEC contribution in this analysis was similar in the whole-genome and exome. This propensity of different mutational signatures to affect coding vs. non-coding regions may result in differences between genome- and exome-based analysis that reflect underlying biology.

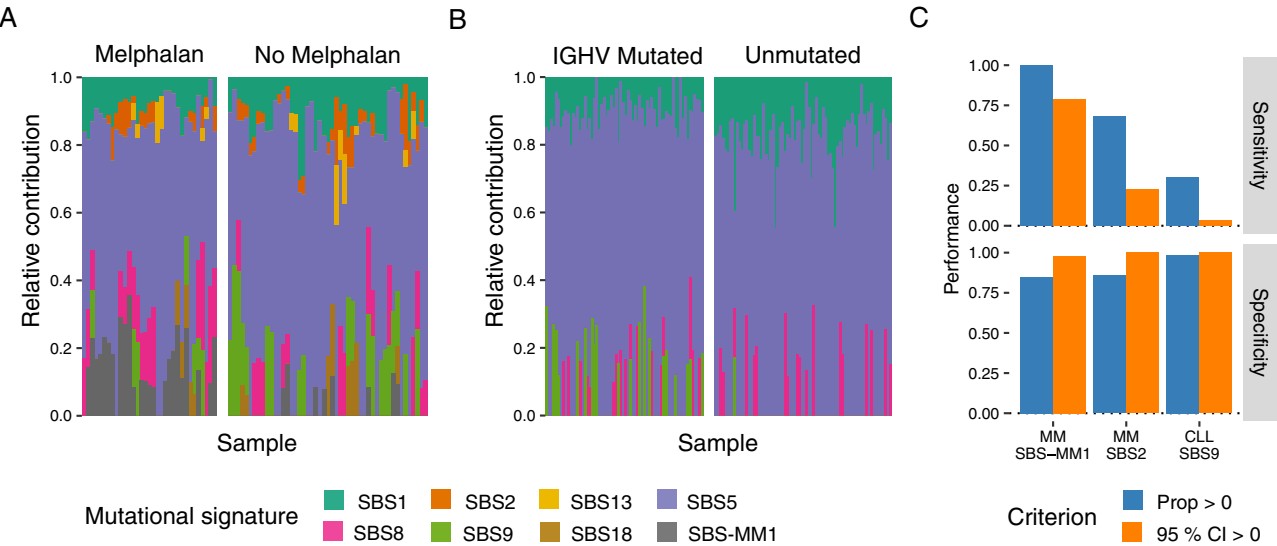

**Fig. 8 Mutational signature fitting applied to whole exomes.** Mutational signature contributions in mutations overlapping the SureSelect V6 + UTR exome capture kit in **A** MM and **B** CLL samples. Mutational signatures from the full whole-genome sequencing data from the same samples are shown in Figs. 2 and 5, respectively. **C** Performance of mutational signature fitting in exome data (shown in **A**, **B**) as compared with the full whole-genome sequencing data as gold standard.

## Discussion

We have shown how mutational signature fitting can be applied to hematological cancers, highlighting the importance of careful interpretation in light of biological knowledge[10]. Our newly released R package *mmsig* is highly specific and provides a range of tools to resolve the presence or absence of a mutational signature in difficult cases. When done properly, signature fitting is a powerful tool that can be implemented immediately in the clinic. The application of mutational signature analysis has the potential to improve prognostic models and define individualized treatment strategies[5,6,23,35–38].

In its essence, a mutational signature is a set of mutation types that show a characteristic pattern of co-occurrence across tumors[3]. Mutational signatures can be defined using the 96-class system of base substitutions in their trinucleotide context but may be applied to any set of features that is biologically meaningful. Although some mutational signatures have important biological implications, they are indirect representations of underlying processes. Distinguishing one signature from another is most straightforward when they involve entirely different parts of the feature-space (i.e., mutational classes), such as the two APOBEC-associated signatures SBS2 and SBS13 in the most recent COSMIC v3.1 reference. Conversely, SBS5 and SBS8 have highly overlapping "flat" profiles; and SBS-MM1 shows some degree of overlap with both SBS5 and SBS9. Highly distinct signatures behave differently from those who are partially overlapping and/or "flat", depending on which mutational signature fitting algorithm and post-processing filters are applied.

Highly characteristic mutational signatures in the latest reference (i.e., COSMIC v3.1) such as SBS2 and SBS13 will stand out over the background even if their relative contribution to the mutational profile is low (<6%). Error-suppression using hard cut-offs, such as the standard 6% threshold imposed by *deconstructSigs*, will be inappropriately strict in these cases. Conversely, *mmsig* will tend to keep these signatures in the profile because removing them will considerably penalize the cosine similarity between the reconstructed mutational profile and the original. The opposite can be said for flat mutational signatures, which may be falsely removed by *mmsig* because all the mutations can be re-classified to another mutational signature with a relatively

low cosine similarity reduction. Thus, *mmsig* inherently requires a larger contribution from indistinctive mutational signatures before they are called. In practice, this dynamic threshold means that *mmsig* is able to call low contribution of APOBEC with high accuracy, while avoiding extensive false positive calls of other (less distinctive) mutational signatures. This could be particularly relevant considering the emerging critical role of APOBEC in predicting myeloma precursor condition progression and MM clinical outcome[23,38–40].

Because different mutational processes occupy largely the same feature space, mutational signature analysis is subject to a degree of uncertainty. We have proposed two objective measures to control uncertainty: estimating 95% CI for the contribution of each mutational signature and checking for transcriptional strand bias typical of signatures associated with transcription-coupled repair. As we have shown empirically, requiring non-zero 95% CI to call a mutational signature as present was highly effective at improving specificity. Relying only on the error-suppression built into *mmsig* resulted in excellent sensitivity and specificity, particularly when applied to catalogs of more than 500 mutations, which is virtually always the case with WGS data. Mutational signature analysis can also be performed on whole-exome sequencing data, with the caveat that the expected signature profile in coding and non-coding regions can be quite different. Targeted capture sequencing panels covering a few megabases often yield too low mutational burdens to allow meaningful mutational signature fitting for individual patients.

In conclusion, we have shown how the novel mutational signature fitting algorithm *mmsig* can be applied to identify biologically and clinically important mutational processes acting in hematological cancers, including chemotherapy-related mutational signatures, APOBEC activity in MM and nc-AID in CLL. The tools and principles outlined here may be applicable in other cancers with a well-characterized mutational signature landscape, taking into account the specific biology and exogenous exposures of each disease. The accuracy of mutational signature fitting, in general, is optimal when applied to catalogs of more than 500 mutations, which can be consistently obtained by WGS. With WGS there is also the opportunity to integrate multiple data types, such as patterns structural variation[31,41,42], to better

pinpoint the mutational machinery and therapeutic suscept-ibilities in a given tumor[5,10,35–37].

## Methods

**Patients and data.** Publicly available SNV data from WGS of patient samples were included in the study: 142 CLL (EGAS00000000092)[28,30], 82 MM from 45 patients (EGAD00001003309 and phs000348.v2.p1)[11,17,18] (7 samples out of the original 89 were removed due to incomplete data on treatment history), 47 AML (phs000178.v1.p1)[43], and two therapy-related AML (EGAD00001005028)[10].

**Mutational signature fitting with *mmsig*.** We developed *mmsig* as a tool for flexible and easily interpretable mutational signature analysis[11]. At its core, *mmsig* takes a set of reference mutational signatures and estimates their contribution in each sample employing an expectation maximization algorithm. For each sample, *mmsig* attempts to reduce the number of features (i.e., reference signatures) used to explain the observed mutational profile. In an iterative process, we reconstructed the 96-class mutational profile for each sample after excluding each reference signature in turn. The least contributing mutational signature was censored for that sample if removal resulted in a cosine similarity reduction of <0.01. This process was subsequently repeated until no reference signatures could be removed without incurring a cosine similarity reduction of more than 0.01. Since SBS1 and SBS5 are known to always be present in all human tumors and normal cells alike, we forced their inclusion in all samples.

The same set of reference mutational signatures can be used for each sample, or a different subset of reference signatures be specified for each sample according to prior knowledge of sample biology or de novo extraction results.

CI were generated by drawing 1000 mutational profiles from the multinomial distribution, where the probability that a mutation belongs to a given class (e.g., C [C>T]G) was equal to the proportion of mutations belonging to that class in the original mutational profile. The number of mutations in each random set of mutations was the same as in the original profile. For each random set of mutations, we repeated the entire mutational signature fitting procedure as described above, finally taking the 2.5th and 97.5th percentile of the estimates for each signature.

Transcriptional strand bias was assessed using a Poisson test ($p < 0.05$). We applied the test both to individual mutation classes independently as well as to the combinations of mutational classes most characteristic of specific mutational signatures, in order to increase power. For SBS-MM1 (melphalan signature) we combined C[C>T]A, G[C>T]A, G[C>T]C, G[C>T]G and G[C>T]T; and for SBS35 (platinum signature) we combined C[C>A]C and C[C>T]C.

*mmsig* is an R package and is available on GitHub: https://github.com/evenrus/mmsig.

**Mutational signature references.** We used the two versions of the COSMIC mutational signature reference (i.e., COSMIC v2 and COSMIC v3.1), with the addition of SBS-MM1 and SBS-HSPC as previously described by our group and others[1,2,10,11,44–46]. The appropriate catalog of reference signatures to include in the analysis for each hematological cancer type was based on de novo signature extraction as previously reported[10,11].

**Comparison of *mmsig* and established mutational signature fitting algorithms.** To benchmark *mmsig* against established tools, we selected the commonly used mutational signature fitting packages *deconstructSigs*[15] and *mutationalPatterns*[16] in R. The same reference signature catalogs were applied for all three algorithms.

Both *mmsig* and *deconstructSigs* have built-in filtering options, which can be altered or turned off entirely. *mmsig* supports dynamic filtering based on cosine similarity as described above. *deconstructSigs* applies a hard threshold of signature contributions in each sample, below which all signatures are removed. *mutationalPatterns* does not have a built-in filtering option, leading us to implement the same filter as applied by *deconstructSigs* as a post-processing step.

**Simulated genomes.** As a basis for simulated MM genomes, we generated an average MM mutational signature profile where the contribution of each signature was set to the median contribution of that signature in MM samples with SBS-MM1. The resulting signature profile consisted of 4.8% SBS1, 5.1% SBS2, 34.1% SBS5, 18.2% SBS8, 18.9% SBS9 and 18.8% SBS-MM1. We then calculated the expected contribution of each of the 96 mutational classes from the weighted sum of each mutational signature reference.

To generate simulated genomes, each simulated mutation was drawn from a multinomial distribution of 96 mutational classes. The probability to draw a mutation of a given class was equal to the relative contribution of that mutational class in the average MM profile.

The sensitivity for each mutational signature was estimated by adding a progressively larger contribution of that signature (1–20%) to a background mutational profile consisting of an MM genome without the signature in question. The background profile was scaled to maintain a constant proportion of each signature relative to the others.

To generate random noise, we generated a "noise signature" independently for each simulation, drawing the contribution of each of the 96 mutational classes from

a poisson distribution[3] with lambda = 2. The weight of the noise signature relative to other signatures was pre-determined (e.g., 5%) and the relative contributions of each mutational signature was scaled accordingly.

**Downsampling of mutational catalogs.** To evaluate the effect of mutation counts on signature fitting performance, we performed downsampling of the full mutation catalogs. Patient samples from MM and CLL were included in this analysis since they both had mutational signatures present in a subset of the cohort where we could define a ground truth regarding the presence or absence of the underlying mutational process (SBS-MM1 in MM and SBS9 (nc-AID) in *IGHV* mutated CLL). We also included SBS2 in the analysis, representing a highly distinctive signature with relatively low abundance in most cases. For each sample, we performed 100 independent draws, without replacement, of 2000 (MM only), 1000, 500, 250, 100. We also generated mutation sets similar to those obtained by whole-exome and targeted sequencing, using BED files of a commonly used exome capture kit (Agilent SureSelect V6 + UTR) and custom targeted capture panels specifically developed for MM[47] and CLL[48,49]. Mutational signature fitting with estimation of 95% CI was performed independently for each mutation set as described above. To estimate sensitivity and specificity for signature detection, we used signature calls from WGS as a gold standard (requiring non-zero 95% CI, and for SBS-MM1, the presence of transcriptional strand bias). Samples with ambiguous signature calls were removed (e.g., if the 95% CI included zero).

**Reporting summary.** Further information on research design is available in the Nature Research Reporting Summary linked to this article.

## Data availability

All the raw data used in the study are already publicly available: EGAS00000000092: 142 CLL. EGAD00001003309 and phs000348.v2.p1: 82 MM from 45 patients, 7 samples out of the original 89 were removed due to incomplete data on treatment history. phs000178.v1.p1: 47 de novo AML. EGAD00001005028: two therapy-related AML.

## Code availability

*mmsig* is an R package and is available on GitHub: https://github.com/evenrus/mmsig.

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

## Acknowledgements

This work is supported by Multiple Myeloma Research Foundation (MMRF), by the Perelman Family Foundation, by Riney Family Multiple Myeloma Research Program Fund, by the Memorial Sloan Kettering Cancer Center NCI Core Grant (P30 CA 008748), and by the Sylvester Comprehensive Cancer Center NCI Core Grant (P30 CA 240139). F.M. is supported by the American Society of Hematology, the International Myeloma Foundation and The Society of Memorial Sloan Kettering Cancer Center. N.B. is funded by the European Research Council under the European Union's Horizon 2020 research and innovation program (grant agreement no. 817997). NA is funded by the European Regional Development Fund and Welsh Government (Ser Cymru programme).

## Author contributions

F.M. designed and supervised the study, collected and analyzed data, and wrote the paper. O.L. supervised the study, collected the data and wrote the paper. E.H.R. designed the study, collected and analyzed data and wrote the paper. F.N., N.A., and B.Z. analyzed data. N.B., X.P., and E.C.

## Competing interests

The authors declare no competing interests.
