## [Peer Review File · Communications Biology]

Reviewers' comments:

Reviewer #1 (Remarks to the Author):

Rustad et al. present mmsig, a software package aiming to improve the specificity of mutational signature extraction by introducing a “dynamic error suppression” procedure, and demonstrate the utility of the algorithm by applying it to hematological cancer datasets. The software fits, similarly to other packages such as deconstructSigs and MutationalPatterns, a set of reference signatures, and implements a simple post-hoc decision rule based on the cosine similarity, rather than relative contribution, to verify the activity of each mutational process. Furthermore the package calculates confidence intervals based on the bootstrap. The application increases the specificity in illustrated use cases, although the overall improvements seem to be moderate in comparison to competing tools.

Major

Even though the stepwise selection based on cosine similarity, one wonders how a formal stepwise model selection for example based on multinomial or dirichlet multinomial likelihood with AIC or BIC compares. This should also resolve the issues arising from low mutation counts.

The authors miss the opportunity to concisely explain the key innovation of their algorithm. I assume it is based on a variant of NMF. Which objective/likelihood is used? Ideally it should be written as a schematic algorithm, for example. Figure 1 of the manuscript illustrates the general idea of mutational signature fitting but not the algorithm itself.

The veracity of CIs and signature selection should also be based on simulations rather than biological plausibility.

Resampling is also employed in sigProfiler if I'm not mistaken. Why didn't the authors compare to this algorithm?

Minor comments

The accompanying text to Fig. 1 defines “reconstruction accuracy” as the cosine similarity between the approximated and the original mutational profile, and then states that algorithms aiming to optimise the latter are often prone to detect false positive signature calls. Although it is clear what the authors mean in this context, it is technically wrong as the signature fitting algorithms don't optimise the cosine similarity, but rather minimise the frobenius norm (squared error) between the mutation counts and their fitted values.

Figure 2 applies mmsig and competing tools to a set of 82 multiple myeloma samples. Although the authors state that all three algorithms show identical results, Fig. 2A reveals upon closer inspection that “No melphalan” samples show lower SBS-MM contributions in mmsig, indicating a lower sensitivity of the algorithm. While Rustad et al. argue that all three algorithms achieve similar reconstruction accuracies (Fig. 2C), they do not provide an indication about the overall fit, i.e. report the likelihood or residuals of the models.

The next section of the manuscript discusses differences between COSMIC releases, and points out that the novel version contains cleaner spectra. The authors highlight differences in the attribution of signature correlates from different COSMIC versions, and emphasize that the attribution of SBS-MM varies if different releases are used. They arrive at the conclusion that their signature selection procedure works best if characteristic spectra are used. The authors take it for granted that the new COSMIC signatures are better, but is it likely that the conclusion is circular in the sense that the newer version was crafted in order to maximise differences between signatures, which produces better fits without necessarily being biologically underpinned.

Figure 6A assesses the performance of mmsig in samples with lower mutational loads, and reveals the bias of the algorithm to preferably attribute mutations to SBS1 and SBS5, which are the only two signatures that cannot be removed from the set of active signatures. Figure 6B shows the sensitivity and specificity of mmsig for different signatures and downsampling degrees, but does not compare the performance to other tools.

Reviewer #2 (Remarks to the Author):

Summary:

The manuscript describes a new process for analysing mutational signatures in a cohort of cancer genomes, and its accompanying software implementation, the R package mmsig.

The authors' previous work (here cited as reference 12) proposed a three-stage workflow for mutational signature analysis: 1) signature extraction, 2) assignment of extracted signatures to corresponding well-defined signatures from a reference set, and 3) fitting those well-defined signatures back to the data. The current paper focuses on stage three.

The fundamental observation in the paper, which builds on that made in the authors' earlier paper, is that mutational signatures are not independent, and consequently methods that fit signatures to data are prone to overfitting and making false positive assignments of signatures to samples that do not possess them, particularly if those signatures are strongly present among other samples in the cohort. The method presented in this paper is intended to be robust to these errors, and is demonstrated on data from multiple myeloma patients, and compared to results obtained from popular packages deconstructSigs and MutationalPatterns.

Impression:

This paper provides a thorough overview of the analysis carried out by the authors, and their software. The potential pitfalls of mutational signature fitting are well observed, and the approach taken in mmsig to avoid overfitting and attributing false positives, although quite simple, is well thought out, appropriate, and by the results presented, effective. I would recommend that this

paper be published without revisions. My only reservation is that the method seems to be tied quite closely to analysis of multiple myeloma in particular, and may not be of wider general interest or application to the cancer research community.

Comments:

1) How specific is mmsig to multiple myeloma data? Given that the software's name includes "mm", it seems specifically targeted to this disease. The argument for using the tool would be more compelling if its improved performance were demonstrated with another cancer type.

2) The default cutoff threshold in deconstructSigs is 6%, i.e. any signature contributing less than 6% of mutations to a sample is removed from that sample. This creates the problem that APOBEC signatures, which are believed to genuinely be present in the samples, are removed because they do not exceed this cutoff. However, deconstructSigs allows the cutoff to be specified per-signature (by providing a vector of cutoffs to the fitting function). This would allow APOBEC signatures, as well as the clocklike signatures SBS1 and 5, to always be retained, by setting a cutoff $\ll 6\%$ for these signatures. If this were done, would deconstructSigs perform as well as mmsig? I recognise that mmsig does not require such manual intervention in setting particular signature weights, which is an advantage to mmsig.

3) This is a comment on the mmsig code, rather than the paper: there appears to be an error in the strand bias test used by mmsig, which is claimed to be the same as that used in MutationalSignatures. Both use R's built in poisson.test function, but MutationalSignatures calls it as below (slightly paraphrasing):

```
poisson.test(c(transcribed, untranscribed), r = 1)
```

While mmsig calls it as:

```
poisson.test(transcribed, untranscribed, r = 1)
```

These give different p-values. I believe the incorrect mmsig formulation is likely to give smaller p-values, which would tend to inflate the statistical significance of the results it reports.

Reviewer #3 (Remarks to the Author):

Title: mmsig: a fitting approach to accurately identify somatic mutational signatures in hematological malignancies

The authors describe a new R package, mmsig, for signature refitting, targeted mostly at hematological malignancies. One of the major features which the authors describe is the dynamic error-suppression procedure they implemented to help detect signatures of low abundance reliably (if they have distinctive features).

The authors include an interesting analysis of the impact on signature refitting when using the new,

cleaner signatures (COSMIC v3.1) as compared to older, more noisy signatures (COSMIC v2) and include further analyses to show the benefit of their approach.

This is very interesting (and well written) work that definitely merits consideration and also publication.

There are, however, a few issues (or rather questions) which arise when reading the manuscript and which the authors should consider:

1) For most of the analyses, the authors used only "the 8 mutational signatures previously identified in MM" (line 115). This is consistent with the focus on hematological malignancies, or maybe "other cancers with a well-characterize mutational signature landscape" as the authors say (line 365).

One wonders however what happens if the method is applied using the full set of mutational signatures (e.g., as an "explanatory" tool for cancer types with less well defined signature contribution).

The error-correction approach of mmsig relies on the distinctiveness of mutational signatures. Including more signatures, however, it is likely that overall the distinctiveness will be less pronounced for individual signatures. How does the error-correction approach cope with that? Does mmsig's error-correction approach still improve the results when taking the full set of signatures?

If this is not the case, it would be good to state this more explicitly. Although the authors do not claimed that mmsig does improve the results when taking the full set of signatures, readers and potential users should be aware. This would in no way diminish the significance of the work, because using mmsig would still be beneficial for cancer types with well-defined subsets of signatures.

If mmsig can still improve the results when using the full signature set, this would be a nice addition. In both cases, additional results for the full set might be shown in the supplementary material.

2) If the noisy COMSICv2 signatures cancel out (or prevent) the positive effect of the mmsig error-correction approach (see Figure 3E), might it not be that mmsig is merely overfitting noise, which is anyway present, by assigning it to the next best (flat) signature(s) when COMSIC v3.1 is used?

Probably not, but to be sure it would be best to perform dedicated tests with simulated data (similar to the used tumor samples but with known signature contribution plus increasing noise levels). Such simulated tumor samples (i.e., their mutational profiles) can be easily generated by multiplying signatures with the desired contributions/exposures, summing them and finally adding some noise.

Such simulated data might also be suited for measuring the performance of mmsig and the other tools.

3) [related to point 2]

Figure 2 suggests that while the mmsig error-correction approach correctly filters out SBS-MM1 from non-treated patients, it may at the same time loose the SBS-MM1 contributed of treated

patients (small contributions present on the left figure, are lost in the right figure).

As you say later (line 215/16), these patients might have had no mutational effect of the treatment such that a melphalan-exposed cell proliferated beyond the limit of detection, but can you be sure that this is indeed the case? With only small contributions you would maybe not have the statistical power to, for example, observe sufficient strand bias (which you use as a sort of control) to know whether SBS-MM1 is truly present or not. This might be another reason to perform some analysis using simulated data, where you can be sure about what to expect.

4) Lines 149-151: "Because SBS1 and SBS5 are known to always be present, we kept them in the final profile no matter what their contribution was to the overall reconstruction accuracy" (also mentioned in the Methods, lines 393-395)

Does that mean this is hard-coded in mmsig? Or did you adjust the result obtained from mmsig accordingly? And if it is hard-coded, can mmsig then be applied to custom signatures sets (e.g., de-novo sets derived from own data), or does it work only with COSMIC v3.1 or COSMIC v2?

5) Lines 192/193: "irrespective of which signature reference was used, only mmsig could avoid false positive SBS-MM1 in patients without prior melphalan exposure".

While I believe that this is indeed the case, you didn't show this for COSMIC v2. (Or maybe I overlooked it?)

Some additional minor comments, suggestions, corrections that you can also safely ignore:

6) Line 87: it would be best to briefly describe with half a sentence what is meant by "inter-bleeding", so that the reader can grasp the main idea without needing to first read the cited paper.

7) Line 519 (caption of Figure 3 B-C): shouldn't it be "SBS1 and SBS5" instead of "SBS1 and SBS18"? If Fig 3C is really for SBS18, then the figure title ("SBS5") is wrong. Given the values on the y-axis, I think this is indeed SBS5 (as you also say later in the caption) and the caption in line 519 is incorrect, not the title of the figure.

8) Figure 3 (and in the supplement): in some places you use "COSMIC 49" and "COSMIC 30" without having them defined; indeed in caption of Figure 3 you define "New" and "Old" instead, which aren't used in the figure. Reading with attention (or knowing the signatures) it is easy to figure out what you mean by "COSMIC 49/30" but I would anyway recommend to explicitly define it when you use it for the first time. Alternatively, you could always use the version number ("COSMIC v3.1") in the figures.

9) In Figure 4b: You should provide a citation/reference for SBS-HSPC as you did for SBS-MM1.

10) Probable error in line 217: "where myeloma cell_s_ are infused"

11) I would suggest to show the most important signatures in a dedicated figure. Remarks such as "... disproportionately affecting flat signatures such as SBS5, while the APOBEC-signatures and SBS1 were relatively spared" (lines 273/4) are not easy to understand for those who aren't familiar with the signatures.

If you prefer not to include such a figure, then I would suggest to move the brief description of the signature profiles (lines 326-333) from the Discussion the introduction or where you first mention the set of signatures you use (Results, line 115)

12) Author contributions: For E.H.R. your list of contributions seem to contain some copy&paste error

Re: *mmsig*: a fitting approach to accurately identify somatic mutational signatures in hematological malignancies

We thank the Reviewers for taking the time to scrutinize our manuscript, providing insightful criticism and suggestions. Please find below our point-by-point response; reviewer comments are in black and our response in blue

=====

REVIEWER COMMENTS

Reviewer #1

Rustad et al. present *mmsig*, a software package aiming to improve the specificity of mutational signature extraction by introducing a “dynamic error suppression” procedure, and demonstrate the utility of the algorithm by applying it to hematological cancer datasets. The software fits, similarly to other packages such as *deconstructSigs* and *MutationalPatterns*, a set of reference signatures, and implements a simple post-hoc decision rule based on the cosine similarity, rather than relative contribution, to verify the activity of each mutational process. Furthermore the package calculates confidence intervals based on the bootstrap. The application increases the specificity in illustrated use cases, although the overall improvements seem to be moderate in comparison to competing tools.

Major

Even though the stepwise selection based on cosine similarity, one wonders how a formal stepwise model selection for example based on multinomial or dirichlet multinomial likelihood with AIC or BIC compares. This should also resolve the issues arising from low mutation counts.

We designed *mmsig* as a fitting algorithm that works on signatures extracted by either NMF or other mutational signature extraction algorithms (e.g. hierarchical Dirichlet process). Creating a more formal model would mean we would have to integrate the two steps. Our design is based on the prior knowledge of which mutational signatures are active in each cancer type. In fact, these signatures have been previously extracted in all these hematological cancers by NMF, *sigprofler* or *hdp*. Similarly to *deconstructsig*, the aim of *mmsig* is not to “de novo” extract signatures, but to accurately quantify their activity (i.e. fitting approach).

The authors miss the opportunity to concisely explain the key innovation of their algorithm. I assume it is based on a variant of NMF. Which objective/likelihood is used? Ideally it should be written as a schematic algorithm, for example. Figure 1 of the manuscript illustrates the general idea of mutational signature fitting but not the algorithm itself.

As mentioned above, this is not a likelihood-based algorithm for mutational signature “de novo” extraction. We are fitting a selection of the signatures extracted by NMF in these three cancers: multiple myeloma, chronic lymphocytic leukemia and acute myeloid leukemia. The overall architecture of the algorithm is described in the methods section (under “*Mutational signature*”).

fitting with mmsig", lines 430-460) and the source code is freely available on github (<https://github.com/evenrus/mmsig>).

The veracity of CIs and signature selection should also be based on simulations rather than biological plausibility.

We have tested our CIs using simulated data, following an approach suggested by Reviewer #3. This confirms that our bootstrapping-based 95 % CIs contain the true value in ~94 % of simulated genomes (lines 451-452). We have also performed additional simulations to validate the performance of *mmsig* and the other algorithms (Results, lines 278-303 and new **Figure 6**).

As pointed out in our previous paper(1) and in the introduction of the manuscript, the *mmsig* mutational signature selection is based on prior knowledge of the mutational signatures involved in that particular type of cancer based on previous studies using de novo signature extraction approaches (e.g., NMF).

Resampling is also employed in *sigProfiler* if I'm not mistaken. Why didn't the authors compare to this algorithm?

The Reviewer correctly pointed out that also *SigProfiler* has within its flow a fitting step which estimates the contribution of a known set of mutational signatures for each individual sample (called: *SigProfilerSingleSample*). We did not include this in our analysis for two main reasons:

- *SigProfilerSingleSample* needs the entire *SigProfilerMatrixGenerator*, *SigProfilerExtractor* and *SigProfilerPlotting* output to run properly. This is obviously critical when we approach a tumor with unknown mutational signature landscape. However, this approach takes also time to be run and it is computational expensive compared to quicker and accurate fitting approaches.
- *SigProfilerSingleSample* is based on the *SigProfiler* output, and this limits the flexibility towards new signatures and possible false positive calls. For example, SBS-MM1 has not been included in the new COSMIC v3.1 yet, and its activity is wrongly splitted across different known signatures.

While *SigProfilerSingleSample* is a critical tool for a comprehensive interpretation of *SigProfiler* output, its flexibility and feasibility as fitting approach are lower compared to *mutationalPatterns*, *mmsig*, and *deconstructSigs*.

Finally, we selected *mutationalPatterns* and *deconstructSigs* for comparison because they are both commonly used packages for mutational signature fitting.

Minor comments

The accompanying text to Fig. 1 defines "reconstruction accuracy" as the cosine similarity between the approximated and the original mutational profile, and then states that algorithms aiming to optimise the latter are often prone to detect false positive signature calls. Although it is clear what the authors mean in this context, it is technically wrong as the signature fitting algorithms don't optimise the cosine similarity, but rather minimise the frobenius norm (squared error) between the mutation counts and their fitted values.

"Optimize" in this setting was intended as a colloquial term. Moreover, although Frobenius norm is minimized in NMF, this is not an established standard for mutational signature fitting algorithms. The inner workings of each algorithm may differ, but we believe that reconstruction accuracy as defined in the paper is still a useful way to understand the principles of how

mutational signature fitting works. Since our article is also intended for a non-statistical readership, we prefer to keep the text as it stands on this issue.

Figure 2 applies *mmsig* and competing tools to a set of 82 multiple myeloma samples. Although the authors state that all three algorithms show identical results, Fig. 2A reveals upon closer inspection that “No melphalan” samples show lower SBS-MM contributions in *mmsig*, indicating a lower sensitivity of the algorithm.

Figure 2A displays the results of mutational signature fitting without any filtering to remove false positives. Several studies have demonstrated that SBS-MM1 is only present after exposure to melphalan, including both in vitro and in vivo evidence(2). Consequently, SBS-MM1 identified by all algorithms in “No melphalan” samples constitute false positives. Although we agree with the Reviewer that this SBS-MM1 signal may be slightly less pronounced in *mmsig*, our data suggests that all three algorithms show similar performance when no filters are applied. These observations were consistent throughout our analyses (**Figures 2C**, and **6C**). We have changed the wording to describe results in Figure 2A from “virtually identical” to “similar”.

After filters have been applied, *mmsig* does show lower sensitivity for SBS-MM1 as compared with the other algorithms, as also shown by our simulated data. That being said, we have also shown that low abundance of SBS-MM1 is ambiguous and should be carefully interpreted in the context of transcriptional strand bias as well as estimation of 95 % CIs. When a subtle signal of SBS-MM1 (i.e., <10%) is called by mutational signature fitting, there is usually insufficient power to confidently ascertain the presence of a true melphalan footprint. These ambiguous cases are sometimes filtered out by *mmsig*. Our aim in designing *mmsig* was to provide additional tools to address whether or not signatures such as SBS-MM1 can be confidently called as present. These aspects are now addressed more explicitly in the manuscript (Results, lines 230-236)

While Rustad et al. argue that all three algorithms achieve similar reconstruction accuracies (Fig. 2C), they do not provide an indication about the overall fit, i.e. report the likelihood or residuals of the models.

mmsig fits signatures identified by “de novo” extraction tools (e.g. NMF) with no likelihood introduced at this stage, so likelihood or residuals cannot be reported. The overall architecture of the algorithm is described in the methods section (under “Mutational signature fitting with *mmsig*”, lines 429-460) and the source code is freely available on github (<https://github.com/evenrus/mmsig>).

The next section of the manuscript discusses differences between COSMIC releases, and points out that the novel version contains cleaner spectra. The authors highlight differences in the attribution of signature correlates from different COSMIC versions, and emphasize that the attribution of SBS-MM varies if different releases are used. They arrive at the conclusion that their signature selection procedure works best if characteristic spectra are used. The authors take it for granted that the new COSMIC signatures are better, but is it likely that the conclusion is circular in the sense that the newer version was crafted in order to maximise differences between signatures, which produces better fits without necessarily being biologically underpinned.

Although we do believe that COSMIC v3.1 better reflects the underlying mutational processes and biology, and there is evidence to support this view(3-5), our discussion of COSMIC v2 and v3.1 are more of a technical nature. In fact, we wanted to highlight how exchanging one signature reference for another may affect the signature analysis in unexpected ways.

Specifically, changing from COSMIC v2 to v3.1 reduces the relative contribution of signatures like SBS1 and SBS2 in many cases below the limit of 6 %, which is applied as a standard filtering threshold by *deconstructSigs*. This will reduce the performance of *deconstructSigs* and similar approaches which apply hard thresholds for filtering based on the relative contribution of signatures. Although the advantage of *mmsig* over the other two algorithms becomes clearer when applying COSMIC v3.1, *mmsig* performs well with both of the references (**Figures 2C-D and 3E**).

Figure 6A assesses the performance of *mmsig* in samples with lower mutational loads, and reveals the bias of the algorithm to preferably attribute mutations to SBS1 and SBS5, which are the only two signatures that cannot be removed from the set of active signatures. Figure 6B shows the sensitivity and specificity of *mmsig* for different signatures and downsampling degrees, but does not compare the performance to other tools.

In the **new Figure 6C** we compare mutational signature estimates from all three algorithms in simulated multiple myeloma genomes with different mutation counts and increasing Poisson noise (as suggested by Reviewer #3). The results indicate that SBS5 is accurately estimated by all algorithms in the absence of noise, with increasing over-estimation by all algorithms as the noise increases. As expected, decreasing the number of mutations also increases variability; this was also observed in real data. When the number of mutations falls below 250, there was a tendency for *mmsig* to over-estimate SBS5, most likely for the reason suggested by the Reviewer. For SBS1, given its highly distinctive profile, the estimates by *mmsig* are highly accurate irrespective of added noise and the number of mutations. This was also the case for SBS2. Furthermore, in the **new Figure 6A-B**, we illustrate the respective sensitivities of all three algorithms to detect SBS2 and SBS-MM1 in simulated data.

Reviewer #2

Summary:

The manuscript describes a new process for analysing mutational signatures in a cohort of cancer genomes, and its accompanying software implementation, the R package *mmsig*.

The authors' previous work (here cited as reference 12) proposed a three-stage workflow for mutational signature analysis: 1) signature extraction, 2) assignment of extracted signatures to corresponding well-defined signatures from a reference set, and 3) fitting those well-defined signatures back to the data. The current paper focuses on stage three.

The fundamental observation in the paper, which builds on that made in the authors' earlier paper, is that mutational signatures are not independent, and consequently methods that fit signatures to data are prone to overfitting and making false positive assignments of signatures to samples that do not possess them, particularly if those signatures are strongly present among other samples in the cohort. The method presented in this paper is intended to be robust to these errors, and is demonstrated on data from multiple myeloma patients, and compared to results obtained from popular packages *deconstructSigs* and *MutationalPatterns*.

Impression:

This paper provides a thorough overview of the analysis carried out by the authors, and their software. The potential pitfalls of mutational signature fitting are well observed, and the approach taken in *mmsig* to avoid overfitting and attributing false positives, although quite

simple, is well thought out, appropriate, and by the results presented, effective. I would recommend that this paper be published without revisions. My only reservation is that the method seems to be tied quite closely to analysis of multiple myeloma in particular, and may not be of wider general interest or application to the cancer research community.

Comments:

1) How specific is *mmsig* to multiple myeloma data? Given that the software's name includes "mm", it seems specifically targeted to this disease. The argument for using the tool would be more compelling if its improved performance were demonstrated with another cancer type.

Mmsig was designed for multiple myeloma and other hematological cancers. As shown in this paper specifically on *mmsig* and our previous paper on the entire mutational signature analysis workflow for hematological cancers(1), similar principles and pitfalls apply across MM, CLL and AML.

2) The default cutoff threshold in *deconstructSigs* is 6%, i.e. any signature contributing less than 6% of mutations to a sample is removed from that sample. This creates the problem that APOBEC signatures, which are believed to genuinely be present in the samples, are removed because they do not exceed this cutoff. However, *deconstructSigs* allows the cutoff to be specified per-signature (by providing a vector of cutoffs to the fitting function). This would allow APOBEC signatures, as well as the clocklike signatures SBS1 and 5, to always be retained, by setting a cutoff $\ll 6\%$ for these signatures. If this were done, would *deconstructSigs* perform as well as *mmsig*? I recognise that *mmsig* does not require such manual intervention in setting particular signature weights, which is an advantage to *mmsig*.

The reviewer provides a feasible alternative approach, while also pointing out what we argue is the main strength of *mmsig*. Despite the increasing focus on mutational signatures, the mutational profiles of tumors are still poorly understood. Each tumor is unique, and the presence of unpredictable biological and technical variation will affect the performance of mutational signature fitting. Using a dynamic approach like cosine similarity for filtering takes this into account, addressing to what extent a signature is necessary to explain the overall profile. There may be settings where the approach employed in *deconstructSigs* and *mutationalPatterns* are best suited, and indeed, we do not suggest that *mmsig* should be the only tool in a mutational signature analysis toolkit. Nonetheless, *mmsig* has certain unique and useful features which we believe warrant its place in the field.

3) This is a comment on the *mmsig* code, rather than the paper: there appears to be an error in the strand bias test used by *mmsig*, which is claimed to be the same as that used in *MutationalSignatures*. Both use R's built in *poisson.test* function, but *MutationalSignatures* calls it as below (slightly paraphrasing):

```
poisson.test(c(transcribed, untranscribed), r = 1)
```

While *mmsig* calls it as:

```
oisson.test(transcribed, untranscribed, r = 1)
```

These give different p-values. I believe the incorrect *mmsig* formulation is likely to give smaller p-values, which would tend to inflate the statistical significance of the results it reports.

We thank the Reviewer for thoroughly reviewing our code and identifying this mistake. It has now been corrected and the appropriate analysis repeated. Particularly for the transcriptional strand bias of platinum-based chemotherapy, the strand bias test is now 100 % sensitive and specific in our AML data.

Reviewer #3

Title: mmsig: a fitting approach to accurately identify somatic mutational signatures in hematological malignancies

The authors describe a new R package, mmsig, for signature refitting, targeted mostly at hematological malignancies. One of the major features which the authors describe is the dynamic error-suppression procedure they implemented to help detect signatures of low abundance reliably (if they have distinctive features).

The authors include an interesting analysis of the impact on signature refitting when using the new, cleaner signatures (COSMIC v3.1) as compared to older, more noisy signatures (COSMIC v2) and include further analyses to show the benefit of their approach.

This is very interesting (and well written) work that definitely merits consideration and also publication.

There are, however, a few issues (or rather questions) which arise when reading the manuscript and which the authors should consider:

1) For most of the analyses, the authors used only "the 8 mutational signatures previously identified in MM" (line 115). This is consistent with the focus on hematological malignancies, or maybe "other cancers with a well-characterize mutational signature landscape" as the authors say (line 365).

One wonders however what happens if the method is applied using the full set of mutational signatures (e.g., as an "explanatory" tool for cancer types with less well defined signature contribution).

The error-correction approach of mmsig relies on the distinctiveness of mutational signatures. Including more signatures, however, it is likely that overall the distinctiveness will be less pronounced for individual signatures. How does the error-correction approach cope with that? Does mmsig's error-correction approach still improve the results when taking the full set of signatures?

If this is not the case, it would be good to state this more explicitly. Although the authors do not claimed that mmsig does improve the results when taking the full set of signatures, readers and potential users should be aware. This would in no way diminish the significance of the work, because using mmsig would still be beneficial for cancer types with well-defined subsets of signatures.

If mmsig can still improve the results when using the full signature set, this would be a nice

addition. In both cases, additional results for the full set might be shown in the supplementary material.

As pointed out, a large number of mutational signatures has now been identified, many of which show considerable overlap in their distinguishing features. Unfortunately, applying *mmsig* without pre-selecting the reference mutational signatures will lead to inclusion of signatures which are not actually present in that tumor type (**Figure R1** below in the rebuttal). The recommended usage is to apply a mutational signature fitting approach such as *mmsig* as the final step of a mutational signature analysis workflow after first identifying which signatures are present in that disease, as we have described previously(1). Thankfully, more and more cancers have now been well-characterized in terms of their mutational signature landscape, including the hematological cancers for which *mmsig* was developed.

2) If the noisy COMSICv2 signatures cancel out (or prevent) the positive effect of the *mmsig* error-correction approach (see Figure 3E), might it not be that *mmsig* is merely overfitting noise, which is anyway present, by assigning it to the next best (flat) signature(s) when COMSIC v3.1 is used?

Probably not, but to be sure it would be best to perform dedicated tests with simulated data (similar to the used tumor samples but with known signature contribution plus increasing noise levels). Such simulated tumor samples (i.e., their mutational profiles) can be easily generated by multiplying signatures with the desired contributions/exposures, summing them and finally

adding some noise.

Such simulated data might also be suited for measuring the performance of *mmsig* and the other tools.

The signatures in COSMIC v2 are not noisy per se. Instead, they contain what we would refer to as background contamination from other signatures, typically from SBS5 and other possible “flat” signatures.

Nonetheless, we recognize that the suggested simulation approach and simulated noise is an excellent idea. In the **new Figure 6C** we compare mutational signature estimates from all three algorithms in simulated multiple myeloma genomes with different mutation counts and increasing Poisson noise. The results indicate that SBS-MM1, SBS5, SBS8 and SBS9 are accurately estimated by all algorithms in the absence of noise, with increasing over-estimation of SBS5 and SBS9 by all algorithms as the noise increases. As expected, decreasing the number of mutations also increases variability; this was also observed in real data. When the number of mutations falls below 250, there was a tendency for *mmsig* to over-estimate SBS5, most likely because this signature was always kept in the final profile irrespective of random variations in the data which may lead to dropout of other signatures. For SBS1, given its highly distinctive profile, the estimates by *mmsig* were highly accurate irrespective of added noise and the number of mutations. This was also the case for SBS2.

3) [related to point 2]

Figure 2 suggests that while the *mmsig* error-correction approach correctly filters out SBS-MM1 from non-treated patients, it may at the same time lose the SBS-MM1 contributed of treated patients (small contributions present on the left figure, are lost in the right figure).

As you say later (line 215/16), these patients might have had no mutational effect of the treatment such that a melphalan-exposed cell proliferated beyond the limit of detection, but can you be sure that this is indeed the case? With only small contributions you would maybe not have the statistical power to, for example, observe sufficient strand bias (which you use as a sort of control) to know whether SBS-MM1 is truly present or not. This might be another reason to perform some analysis using simulated data, where you can be sure about what to expect.

The reviewer raises an important question regarding the power to confidently ascertain the presence or absence of SBS-MM1.

In simulated data we show that all three algorithms have virtually identical sensitivity for SBS-MM1 in the absence of filtering, but *mmsig* showed lower sensitivity after filtering (in contrast to distinctive signatures such as SBS2 where *mmsig* has superior sensitivity) (**new Figure 6A-B**). That being said, we have also shown that low abundance of SBS-MM1 is ambiguous and should be interpreted in the context of transcriptional strand bias as well as estimation of 95 % CIs. When a subtle signal of SBS-MM1 (i.e., <10%) is called by mutational signature fitting, there is usually insufficient power to confidently ascertain the presence of a true melphalan footprint. These ambiguous cases are sometimes filtered out by *mmsig*. Our aim in designing *mmsig* was to provide additional tools to address whether or not signatures such as SBS-MM1 can be confidently called as present. We prefer to be conservative in these cases and tolerate false negatives to a greater degree than false positives.

4) Lines 149-151: "Because SBS1 and SBS5 are known to always be present, we kept them in

the final profile no matter what their contribution was to the overall reconstruction accuracy" (also mentioned in the Methods, lines 393-395)

Does that mean this is hard-coded in *mmsig*? Or did you adjust the result obtained from *mmsig* accordingly? And if it is hard-coded, can *mmsig* then be applied to custom signatures sets (e.g., de-novo sets derived from own data), or does it work only with COSMIC v3.1 or COSMIC v2?

In the previous version of *mmsig*, this was indeed hard-coded. SBS5 and SBS1 has been identified ubiquitously across the animal kingdom and we reasoned that they should always be present in a cancer genome. Although we believe that this assumption still holds, we also realize there may be other applications of *mmsig* where such behavior is not desired. We have therefore changed the *mmsig* package such that signatures to keep in the profile can be specified at the function call. Always keeping SBS1 and SBS5 in the final profile is still the default option.

5) Lines 192/193: "irrespective of which signature reference was used, only *mmsig* could avoid false positive SBS-MM1 in patients without prior melphalan exposure".

While I believe that this is indeed the case, you didn't show this for COSMIC v2. (Or maybe I overlooked it?)

This was indeed the case also for COSMIC v2. We have included these data as a new **Figure S3**

Some additional minor comments, suggestions, corrections that you can also safely ignore:

6) Line 87: it would be best to briefly describe with half a sentence what is meant by "inter-bleeding", so that the reader can grasp the main idea without needing to first read the cited paper.

We thank the Reviewer for this suggestion – a brief explanation has been added.

7) Line 519 (caption of Figure 3 B-C): shouldn't it be "SBS1 and SBS5" instead of "SBS1 and SBS18"? If Fig 3C is really for SBS18, then the figure title ("SBS5") is wrong. Given the values on the y-axis, I think this is indeed SBS5 (as you also say later in the caption) and the caption in line 519 is incorrect, not the title of the figure.

Corrected – it should indeed have said SBS5.

8) Figure 3 (and in the supplement): in some places you use "COSMIC 49" and "COSMIC 30" without having them defined; indeed in caption of Figure 3 you define "New" and "Old" instead, which aren't used in the figure. Reading with attention (or knowing the signatures) it is easy to figure out what you mean by "COSMIC 49/30" but I would anyway recommend to explicitly define it when you use it for the first time. Alternatively, you could always use the version number ("COSMIC v3.1") in the figures.

We have changed terminology to use v2 and v3 consistently.

9) In Figure 4b: You should provide a citation/reference for SBS-HSPC as you did for SBS-MM1.

References for SBS-HSPC has been added in the methods section under "Mutational signature references".

10) Probable error in line 217: "where myeloma cell_s_ are infused"

Corrected

11) I would suggest to show the most important signatures in a dedicated figure. Remarks such as "... disproportionately affecting flat signatures such as SBS5, while the APOBEC-signatures and SBS1 were relatively spared" (lines 273/4) are not easy to understand for those who aren't familiar with the signatures.

If you prefer not to include such a figure, then I would suggest to move the brief description of the signature profiles (lines 326-333) from the Discussion the introduction or where you first mention the set of signatures you use (Results, line 115)

The main signatures are included in Figure 1B. Comparison of COSMIC v2 and v3.1 is shown in figure S2 for SBS1 and SBS2 to illustrate the main differences.

12) Author contributions: For E.H.R. your list of contributions seem to contain some copy&paste error

Corrected.

Rebuttal references

1. Maura F, Degasperi A, Nadeu F, Leongamornlert D, Davies H, Moore L, et al. A practical guide for mutational signature analysis in hematological malignancies. *Nature communications*. 2019;10(1):2969.
2. Rustad EH, Yellapantula V, Leongamornlert D, Bolli N, Ledergor G, Nadeu F, et al. Timing the initiation of multiple myeloma. *Nature communications*. 2020;11(1):1917.
3. Petljak M, Alexandrov LB, Brammell JS, Price S, Wedge DC, Grossmann S, et al. Characterizing Mutational Signatures in Human Cancer Cell Lines Reveals Episodic APOBEC Mutagenesis. *Cell*. 2019;176(6):1282-94.e20.
4. Sanders MA, Chew E, Flensburg C, Zeilemaker A, Miller SE, Al Hinai AS, et al. MBD4 guards against methylation damage and germ line deficiency predisposes to clonal hematopoiesis and early-onset AML. *Blood*. 2018;132(14):1526-34.
5. Alexandrov LB, Kim J, Haradhvala NJ, Huang MN, Tian Ng AW, Wu Y, et al. The repertoire of mutational signatures in human cancer. *Nature*. 2020;578(7793):94-101.

REVIEWERS' COMMENTS:

Reviewer #1 (Remarks to the Author):

I'd like to thank the authors for their revisions.

It's reassuring to see that the coverage of CIs is as expected (comment 2)

I would have liked to see a comparison to a statistically principled selection criterion (which is for sure possible also for the simple regression task of fitting existing signatures to data, unlike what the authors say), but it's ultimately up to the authors to discuss their rationale. The tools seems to work.

I have no further comments.

Reviewer #2 (Remarks to the Author):

Summary

The manuscript describes a new process for analysing mutational signatures in a cohort of cancer genomes, and its accompanying software implementation, the R package mmsig. This manuscript has been revised since I last saw it. There is now a new section that assesses the performance of mmsig and other software on simulated data.

Impression

I had a favourable opinion of the paper prior to the new revision, except for a few minor comments. These have been resolved to my satisfaction. I now have some comments about the new simulation sections. These mainly concern clarity.

Comments

Concerning confidence intervals (lines 446-453). The authors describe a bootstrap procedure for constructing a confidence interval around their signature contribution estimates. Then they state "Using simulated genomes, we confirmed that the 95% confidence interval contained the true value ~94% of the time.". This is kind of a throw-away line: I don't think this simulation-derived result is presented anywhere else in the paper (unless I missed it). It also reads here as if the "true" value being referred to is the estimate made by mmsig after analysis of the data - this would be highly overconfident! I assume instead that "true" is whatever predetermined signature contribution was used to generate the simulated data, and that in 94% of the simulation results, the 95%CI includes those input values. This should be mentioned in the Results section, rather than here in the Methods.

The section describing the generation of simulated genomes (lines 481-489) is very brief, and I would benefit from a more thorough explanation. The way I interpret it is that some set of signatures is chosen in advance, and a set of mixing weights (signature contributions) is chosen for each simulated sample. The signatures are multiplied by the weights and normalised to generate a table of probabilities of seeing each mutation category in each sample. Each simulated data sample is then generated by drawing from a multinomial distribution parameterised by this probability (and some noise is added for good measure). The aim then for a fitting method is to recover the signature contributions used to generate the data. Is this correct?

Lines 486-7: What does it mean that a signature is removed from the background profile, and adding a progressively larger contribution. Is the signature removed prior to generating the simulated data, or is it done as a post-processing step? Are the other signatures kept in their original proportions relative to each other?

Line 489: Poisson noise: is the same lambda parameter used for each mutation category, to generate flat noise?

Concerning sensitivity analysis (lines 279-304). Is it concerning that mmsig is much less sensitive than the other methods to SBS-MM1, given that mmsig is designed for use on haematological cancers that might be expected to show signs of exposure to SBS-MM1?

Minor comments

In the new figure 6, in panels A and B, it might be better to include separate plots for the filtered and unfiltered results, as in the current layout the results are superimposed to an extent that you can't really read them.

Reviewer #3 (Remarks to the Author):

Overall, nice work.

Re: *mmsig*: a fitting approach to accurately identify somatic mutational signatures in hematological malignancies

We thank the Reviewers for considering our revised manuscript. Please find below our point-by-point response; Reviewer comments are in black and our response in blue

=====

REVIEWER COMMENTS

Reviewer #1:

I'd like to thank the authors for their revisions.

It's reassuring to see that the coverage of CIs is as expected (comment 2)

I would have liked to see a comparison to a statistically principled selection criterion (which is for sure possible also for the simple regression task of fitting existing signatures to data, unlike what the authors say), but it's ultimately up to the authors to discuss their rationale. The tools seems to work.

I have no further comments.

We thank the Reviewer for their feedback.

Reviewer #2:

Summary

The manuscript describes a new process for analysing mutational signatures in a cohort of cancer genomes, and its accompanying software implementation, the R package *mmsig*. This manuscript has been revised since I last saw it. There is now a new section that assesses the performance of *mmsig* and other software on simulated data.

Impression

I had a favourable opinion of the paper prior to the new revision, except for a few minor comments. These have been resolved to my satisfaction. I now have some comments about the new simulation sections. These mainly concern clarity.

Comments

Concerning confidence intervals (lines 446-453). The authors describe a bootstrap procedure for constructing a confidence interval around their signature contribution estimates. Then they state "Using simulated genomes, we confirmed that the 95% confidence interval contained the true value ~94% of the time.". This is kind of a throw-away line: I don't think this simulation-derived result is presented anywhere else in the paper (unless I missed it). It also reads here as if the "true" value being referred to is the estimate made by *mmsig* after analysis of the data - this would be highly overconfident! I assume instead that "true" is whatever predetermined signature contribution was used to generate the simulated data, and that in 94% of the

simulation results, the 95%CI includes those input values. This should be mentioned in the Results section, rather than here in the Methods.

We thank the Reviewer for this suggestion. Indeed, the “true” value in this analysis refers to the predetermined signature contribution. This is now clearly stated in the Results section (page 14, lines 299-301).

The section describing the generation of simulated genomes (lines 481-489) is very brief, and I would benefit from a more thorough explanation. The way I interpret it is that some set of signatures is chosen in advance, and a set of mixing weights (signature contributions) is chosen for each simulated sample. The signatures are multiplied by the weights and normalised to generate a table of probabilities of seeing each mutation category in each sample. Each simulated data sample is then generated by drawing from a multinomial distribution parameterised by this probability (and some noise is added for good measure). The aim then for a fitting method is to recover the signature contributions used to generate the data. Is this correct?

This is correct. For simplicity of interpretation, we did not simulate specific samples, but an average multiple myeloma genome signature profile, based on the average signature weights across patients. We have added additional details in the methods section regarding the simulated genomes, also pertaining to the questions below (page 22-23, lines 478-497).

Lines 486-7: What does it mean that a signature is removed from the background profile, and adding a progressively larger contribution. Is the signature removed prior to generating the simulated data, or is it done as a post-processing step? Are the other signatures kept in their original proportions relative to each other?

In these simulations, the weight of the variable signature was changed prior to generating simulated data. Regarding the other signatures, the Reviewer is right that the other signatures were kept in their original proportions relative to each other. This is now explained in more detail in the manuscript (page 22-23, lines 478-497).

Line 489: Poisson noise: is the same lambda parameter used for each mutation category, to generate flat noise?

We randomly generated a “noise signature” independently for each simulation, drawing the contribution of each of the 96 mutational classes from a poisson distribution with $\lambda = 2$. The weight of the noise signature relative to other signatures was pre-determined (e.g., 5 %). The relative contributions of each mutational signature were scaled to accommodate the noise. This is now explained in more detail in the manuscript (page 22-23, lines 478-497).

Concerning sensitivity analysis (lines 279-304). Is it concerning that *mmsig* is much less sensitive than the other methods to SBS-MM1, given that *mmsig* is designed for use on haematological cancers that might be expected to show signs of exposure to SBS-MM1?

The Reviewer is correct to point out that lower contributions of SBS-MM1 (e.g., <10 %) are often missed by *mmsig*. However, these cases would have been difficult to interpret even if the signature was assigned as present, as we address in the manuscript section on **Resolving uncertainty in mutational signature fitting** (page 10, lines 204-234). In such cases, the mutations assigned to SBS-MM1 could often also have been assigned to another signature without a substantial drop in reconstruction accuracy (hence why it is missed by *mmsig*). In

these cases, we lean on other evidence such as transcriptional strand bias and estimating the 95 % CI of the signature estimate. We have shown how less stringent methods, such as *deconstructSigs* with standard filters, result in false positive signature calls which are avoided by *mmsig*. At the end of the day, we believe the selection of an appropriate tool will depend on the clinical or biological problem at hand.

Minor comments

In the new figure 6, in panels A and B, it might be better to include separate plots for the filtered and unfiltered results, as in the current layout the results are superimposed to an extent that you can't really read them.

We thank the Reviewer for pointing out the challenge in interpreting these figures. Unfortunately, the main graphical problem is that all the unfiltered data are almost identical, as are the filtered results from *MutationalPatterns* and *deconstructSigs*. Thus, changing the figure as suggested would not really improve the interpretability (as shown below). We could in theory have shifted the curves slightly in order to avoid overlap, but this would give the impression that the methods perform differently, which is not the case. Instead, we have re-written the figure legend to emphasize which curves are superimposed (page 29, lines 619-622).

Reviewer #3:

Overall, nice work.

We thank the Reviewer for their feedback.